# Structure-based prediction and characterization of photo-crosslinking in native protein–RNA complexes

Huijuan Feng ®[1,2,3,4], Xiang-Jun Lu[5], Suvrajit Maji ®[2,3,4], Linxi Liu[6,7], Dmytro Ustianenko[2,3,4], Noam D. Rudnick[3,8] & Chaolin Zhang ®[2,3,4] ✉

UV-crosslinking of protein and RNA in direct contacts has been widely used to study protein-RNA complexes while our understanding of the photo-crosslinking mechanisms remains poor. This knowledge gap is due to the challenge of precisely mapping the crosslink sites in protein and RNA simultaneously in their native sequence and structural contexts. Here we systematically analyze protein-RNA interactions and photo-crosslinking by bridging crosslinked nucleotides and amino acids mapped using different assays with protein-RNA complex structures. We developed a computational method PxR3D-map which reliably predicts crosslink sites using structural information characterizing protein-RNA interaction interfaces. Analysis of the informative features revealed that photo-crosslinking is facilitated by base stacking with not only aromatic residues, but also dipeptide bonds that involve glycine, and distinct mechanisms are utilized by different RNA-binding domains. Our work suggests protein-RNA photo-crosslinking is highly selective in the cellular environment, which can guide data interpretation and further technology development for UV-crosslinking-based assays.

Control of RNA metabolism, which is a critical component of gene expression regulation, relies on specific sequence or structural elements embedded in transcripts that are recognized by RNA-binding proteins (RBPs)[1,2]. Such regulatory elements are frequently short (3–7 nucleotides) and degenerate, and protein–RNA interactions are highly dynamic, so it remains a challenge to understand how RBPs specifically recognize their targets[3–5]. A widely used approach to investigate protein–RNA interactions is to crosslink protein and RNA in direct contacts using UV light, which induces covalently linked conjugates between the interacting amino acids and nucleotides[6–8]. UV crosslinking can be performed for tissues or cultured cells and crosslinked protein–RNA complexes can then be analyzed to determine both the

RNA and protein components using different strategies. For example, crosslinking and immunoprecipitation (CLIP) is now the *de facto* standard method to isolate RNA fragments crosslinked to a particular RBP of interest followed by deep sequencing to map RBP binding footprints on a genome-wide scale[9–11]. Alternatively, RBPs can be pulled down through crosslinked RNA to recover their identities using RNA-interactome capture[12–14]. These efforts have greatly expanded the list of RBPs and our understanding of how they contribute to RNA regulation.

Despite the wide applications of UV crosslinking-based assays, the biophysical basis of protein–RNA crosslinking is currently poorly understood, especially for native complexes in vivo or in cellular

[1]Department of Biostatistics and Computational Biology, School of Life Sciences, Fudan University, Shanghai 200438, China. [2]Department of Systems Biology, Columbia University, New York, NY 10032, USA. [3]Department of Biochemistry and Molecular Biophysics, Columbia University, New York, NY 10032, USA. [4]Center for Motor Neuron Biology and Disease, Columbia University, New York, NY 10032, USA. [5]Department of Biological Sciences, Columbia University, New York, NY 10027, USA. [6]Department of Statistics, Columbia University, New York, NY 10027, USA. [7]Present address: Department of Statistics, University of Pittsburgh, Pittsburgh, PA 15260, USA. [8]Present address: Wilmer Eye Institute, Johns Hopkins University, Baltimore, MD 21287, USA. ✉e-mail: cz2294@columbia.edu

contexts. Earlier studies using in vitro crosslinking model systems that involve single amino acids (or dipeptides) and homopolynucleotide chains suggest that each of the RNA bases is capable of forming conjugates with a variety of amino acids or peptides[15,16]. However, these studies provide limited information about photo-crosslinking of macromolecular complexes in cells. To improve the resolution of protein−RNA interaction mapping, approaches have been developed to pinpoint the exact crosslinked nucleotide or amino acid at single-residue resolution, taking advantage of the covalently linked amino acid-nucleotide adducts. Previously, we developed computational methods to map the exact crosslink sites in RNA through analysis of crosslink-induced mutation sites (CIMS) or truncation sites (CITS) using CLIP data[17–19]. CIMS and CITS provide signatures of protein−RNA crosslinking introduced by the amino acid-RNA adducts interfering with the reverse transcription process. The RNA-interactome capture workflow has also been refined to map the crosslinked amino acids by considering the mass shift caused by the RNA moieties conjugated to the crosslinked peptides that are subject to mass-spectrometry analysis[20–22]. However, to our knowledge, no current technologies can map crosslinked amino acid and nucleotide simultaneously in native protein−RNA complexes, although such efforts have been made for individual protein−RNA complexes reconstituted in vitro in a few cases[23–25]. Consequently, it remains unclear how selective photo-crosslinking is or whether particular types of amino acid-nucleotide contacts are required for photo-adduct formation. This knowledge is highly relevant for understanding protein−RNA complex structures and interpreting data generated by UV crosslinking-based assays.

We reasoned that the sequence and structural contexts of protein−RNA crosslink sites can be deduced by integrating the crosslinked nucleotides in RNA inferred by CIMS and CITS analysis and crosslinked amino acids identified in RNA-interactome capture with experimentally determined protein−RNA complex structures. In this work, we developed a computational method named PxR3D-map for this purpose. PxR3D-map systematically analyzes various structural features associated with each nucleotide and amino acid in direct contact within 3D protein-RNA complex structures, which are then used to classify crosslinked vs. non-crosslinked nucleotides, as well as crosslinked vs. non-crosslinked amino acids. More importantly, PxR3D-map ranks structural features based on their importance for classification, which provides mechanistic insights into the biophysical basis of protein−RNA photo-crosslinking and adduct formation in their native complexes.

## Results

### PxR3D-map method overview

The central idea of PxR3D-map is to overlay UV crosslink sites in RNA nucleotides obtained from CLIP data and crosslinked amino acids in proteins obtained from RNA-interactome capture onto experimentally determined protein-RNA complex structures, as deposited in the Protein Data Bank (PDB)[26]. The use of PxR3D-map to integrate CLIP and structure data is illustrated in Fig. 1a. For each protein−RNA complex, we determine the crosslinked nucleotide(s) in the RNA ligand by searching for CIMS/CITS in all instances of this sequence in the transcriptome. We then examine how the crosslinked nucleotide(s) interact with amino acids in the 3D complex structure to find structural features uniquely associated with these nucleotides as compared to non-crosslinked nucleotides. To automate this process, we search for various structural features associated with each nucleotide in the RNA ligand using programs DSSR and SNAP in the 3DNA software suite[27,28]. In total, 15 groups of structural features are extracted to annotate each nucleotide, including RNA nucleotide conformation (e.g., base conformation and sugar puckering) and RNA-secondary structural features (e.g., single vs. double stranded region). The types of protein−RNA contacts considered include hydrogen bonds, planar amino-acid side-chain base stacking (π-stacking), and planar amino-

acid base pairing (pseudo pairing)[29]. To facilitate machine learning-based classification of each nucleotide's crosslinking status, the list of features is tabularized by aggregating the same type of contacts for each unique amino acid (e.g., 20 features that summarize the number of hydrogen bonds with each of the 20 amino acids). Additional features are also included by aggregating amino acids with similar properties (six categories: polar, positive, negative, hydrophobic, aromatic and aliphatic). This results in a total of 246 structural features associated with each nucleotide (Supplementary Data 1). Nucleotides from all protein−RNA complexes with both CLIP data and PDB structures are then compiled together, and the crosslinking status of each nucleotide is classified by its structural features using a random forest model, which also ranks features based on their importance for classification. Similarly, for protein−RNA complexes with crosslinked amino acids identified by RNA-interactome capture, the same strategy is applied to predict the crosslinking status of each amino acid based on its associated structural features (Supplementary Data 2).

To test the feasibility of this strategy, we first examined several examples that 1) represent different types of RNA-binding domains (RBDs) that recognize distinct RNA sequence motifs; and 2) have unambiguously determined crosslink sites in the RNA and protein. We previously demonstrated that RBFOX, which binds the UGCAUG motif through an RNA-recognition motif (RRM), crosslinks to guanines G2 and G6 in the motif sequence by CIMS and CITS analysis[18]. Not surprisingly, G2 and G6 were identified as the predominant crosslink sites when we performed a similar search using the RNA ligand, the heptamer UGCAUGU, which was used to determine RBFOX1 RRM-RNA complex structure (PDB accession: 2ERR[30]; Fig. 1b). These two nucleotides form base-stacking interactions with two phenylalanines, F126 and F160, respectively, which were recently identified as major crosslinked amino acids[21,23,31]. In addition, R118 was also found to be crosslinked in a recent study[21], and this amino acid forms two hydrogen bonds with G6. In a second example, we checked the complex formed by LIN28 and precursor microRNA pre-let7-f1 (PDB accession: 3TS0; ref. 32), with a particular focus on the region interacting with the cold shock domain (CSD). We previously determined that LIN28 CSD recognizes a UGAU motif, with crosslinking at the last uridine[33]. This was confirmed when we searched crosslink sites using the CSD binding region sequence (UAUGAUAC) of pre-let7-f1. The crosslinked uridine stacks with phenylalanine F55, which was also experimentally validated as the crosslinked amino acid[24] (Fig. 1c). In the last example, we examined PUM1, which recognizes an 8-mer motif UGUANAUA through eight Pumilio RNA-binding repeats (Fig. 1d). When we searched CLIP data using the RNA ligand in complex structure (PDB accession: 1M8Y[34]), we found crosslinking to the first uridine (U1) of the motif and also another uridine immediately upstream (U-1). In the crystal structure, U1 stacks with a tyrosine (Y1123); U-1 does not directly contact the protein in the structure, but several amino acids in the vicinity (K1153 and Y1154) were found to be crosslinked to RNA[21], and they may interact with U-1 in cells. These examples demonstrate a striking degree of selectivity of crosslink sites between protein and RNA, indicating requirements of certain structural features of protein−RNA contacts to induce photo-crosslinking in the cellular environment.

### Distinct structural features associated with crosslinked nucleotides

Among RBPs with both high-resolution protein−RNA complex structures deposited in PDB and in-depth CLIP data available, we were able to infer crosslink sites in the RNA ligand unambiguously for 29 non-redundant protein−RNA complexes representing 25 RBPs (Supplementary Data 3; Methods). These complexes have a total of 214 nucleotides in the RNA ligands directly contacting the proteins, including 43 nucleotides that were defined as crosslinked nucleotides and the remaining 171 as non-crosslinked nucleotides. Structural

features associated with each nucleotide were extracted as described above (Supplementary Data 4).

We first performed a statistical comparison of crosslinked vs. non-crosslinked nucleotides interacting with protein by examining individual sequence and structural features. We observed statistical differences in base composition between the two groups (p = 0.0013, chi-squared test), with enrichment of guanine in the crosslinked group; uridine, which is known to be susceptible to UV-crosslinking[35,36], is similarly enriched in the two groups (Fig. 2a).

This pattern appears to be driven mainly by crosslinking to RRMs (Fig. 2b).

RNA nucleotide conformation has been implicated to play a role in protein–RNA recognition[37]. Interestingly, the crosslinked nucleotides favor the C2'-endo conformation in their sugar puckers, while the non-crosslinked nucleotides show similar percentages in C2'-endo or C3'-endo conformation (odds ratio=4.1, p = 0.005, Fisher's exact test; Fig. 2c). In addition, the enrichment of the C2'-endo conformation is particularly prominent for crosslinked guanines and uridines (Fig. 2d).

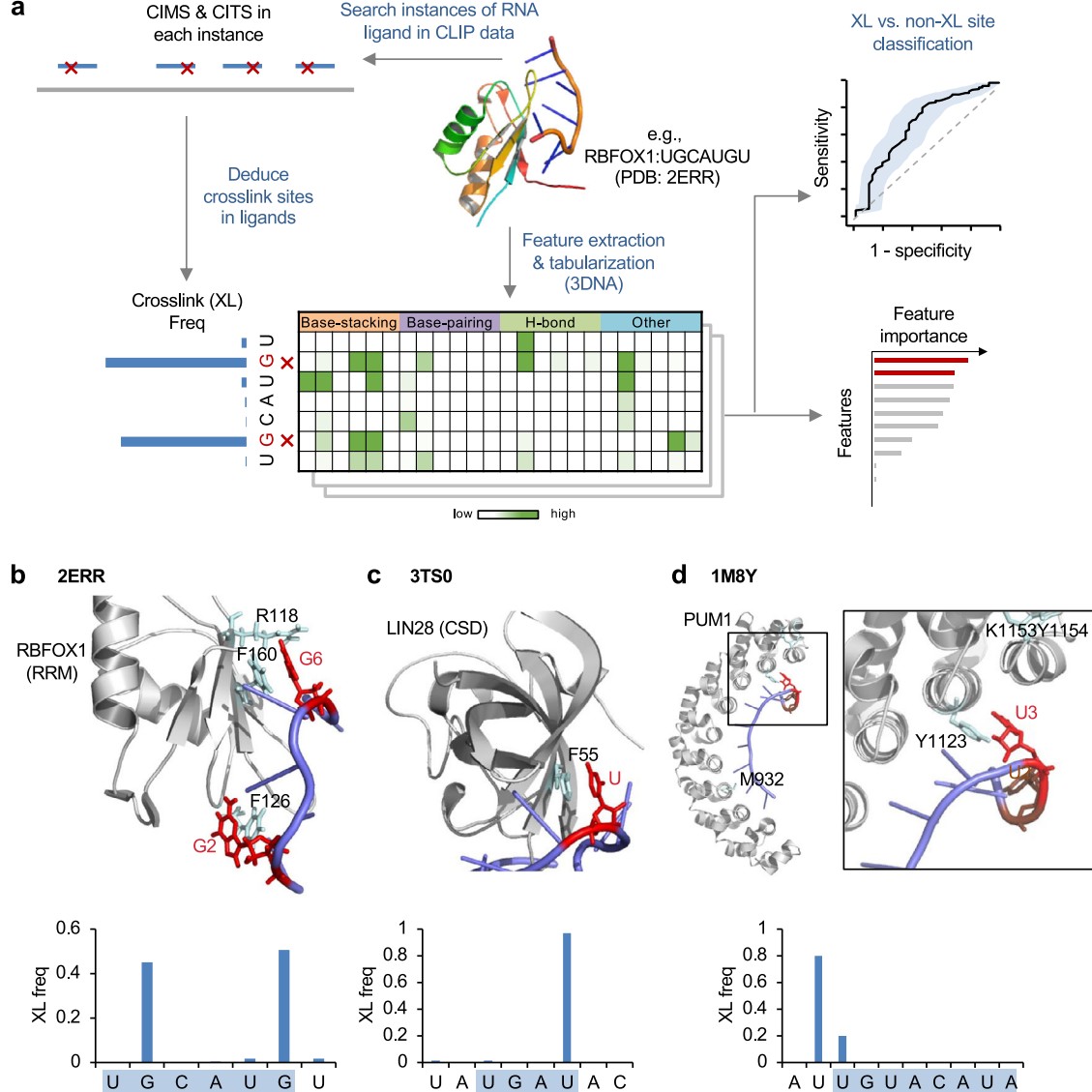

**Fig. 1 | Overview of PxR3D-map to predict photo-crosslinking in native protein–RNA complexes. a** Schematic of PxR3D-map to predict crosslinked nucleotides in RNA using structural features. For structurally resolved protein–RNA complexes, the crosslink sites in the RNA ligand are determined by searching CIMS and CITS in instances of this sequence in CLIP data. Structural features associated with the crosslink sites including how these nucleotides contact amino acids in the complex are then examined. Specifically, the protein–RNA complex structure is analyzed by DSSR and SNAP in the 3DNA software suite to automatically extract various structural features including RNA nucleotide conformation, secondary structures and various types of nucleotide-amino acid contacts including planar amino acid sidechain-base stacking (BS), pseudo base pairing (BP), and different types of hydrogen bonds (H-bond). These structural features are tabulated based on their association with each nucleotide in the RNA ligand (e.g., how many base stacking interactions of a nucleotide with each of the 20 amino acids). These

features are used to predict the crosslinking status of each nucleotide by training a random forest model, which is also used to rank feature importance for their contribution to classification. **b–d** Three examples of protein–RNA complexes. For each complex, the bar plot shows the crosslinking frequency of each nucleotide position, and the major crosslinked nucleotides are indicated at the bottom. The crosslinking frequency was determined by searching all instances of the RNA ligand sequence in CLIP data and counting crosslink sites in different positions of these instances as determined by CIMS and CITS analysis. The structure of the protein–RNA complex illustrated using PyMOL[67] is shown at the top, with RNA in pale blue and protein in gray cartoons. The crosslinked nucleotides (red) in RNA and the crosslinked amino acids (cyan) in protein are shown in sticks to highlight the nucleotide-amino acid contacts. **b** RBFOX1 RRM in complex with UGCAUGU. **c** LIN28 in complex with pre-let7-f1. **d** PUM1 in complex with AUUGUACAUA. The PDB accession code of each structure is indicated.

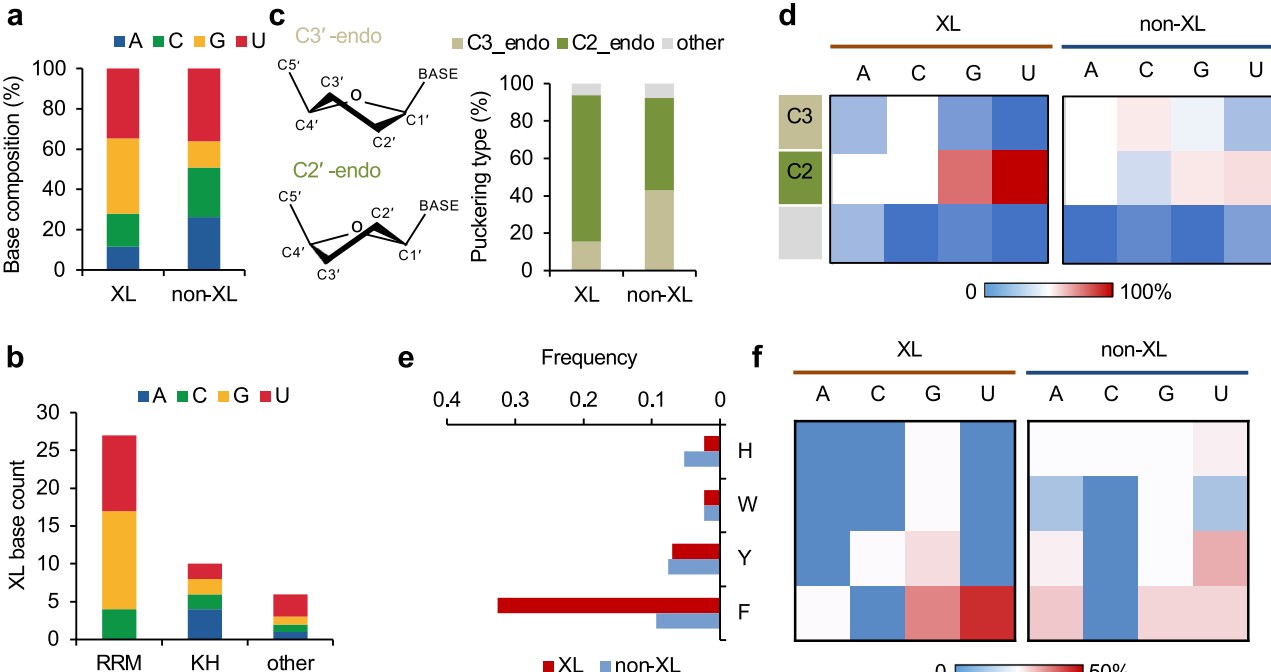

**Fig. 2 | Comparison of crosslinked and non-crosslinked nucleotides using associated structural features.** Only nucleotides in direct contact with amino acids were included in the analysis. **a** Base composition of crosslinked (XL) vs. non-crosslinked (non-XL) nucleotides. **b** Base composition of crosslinked nucleotides for RRMs, KH-domains and other types of RBDs. **c** Distribution of sugar puckering types for crosslinked vs. non-crosslinked nucleotides. The schematics of C2′-and C3′-endo conformations are shown on the left. **d** Similar to (**c**), but shown for each nucleotide base separately using heatmaps. **e** Distribution of base stacking with aromatic amino acids for crosslinked vs. non-crosslinked nucleotides. **f** Similar to (**e**), but shown for each nucleotide base separately using heatmaps.

We also observed that nucleotides in direct contact with protein more frequently adopt the anti-conformation rather than the syn-conformation in the base, and the preference is greater for pyrimidines (Supplementary Fig. 1a,b), which is consistent with a previous study[37]. However, there are no notable differences in base conformation between crosslinked and non-crosslinked nucleotides.

We next examined amino acids in direct contact with crosslinked vs. non-crosslinked nucleotides but did not observe notable differences in amino acid composition between the two groups (Supplementary Fig. 1c), suggesting that the amino acid identity alone does not determine the specificity of crosslinking. In contrast, when we compared the type of protein–RNA contacts, we found a significant enrichment of base stacking with phenylalanine for crosslinked nucleotides as compared to the non-crosslinked nucleotides (0.33 vs. 0.09 events per site, p = 9.5e-4, Binomial test; Fig. 2e, f). From this analysis, we did not observe notable differences in other types of amino acid-nucleotide contacts, such as hydrogen bonds (Supplementary Fig. 1d), despite their importance for determining both specificity and affinity of protein–RNA interactions.

**Prediction of crosslinked RNA nucleotides based on structural features**

To assess the contribution of different structural features and their combinations more systematically to photo-crosslinking in protein–RNA complexes, we applied random forest-based classification models to predict the crosslinked vs. non-crosslinked nucleotides using structural features. For this analysis, the performance of the model was evaluated by a 10-fold cross-validation. Notably, structural features are clearly predictive of crosslinked vs. non-crosslinked nucleotides with an AUC (area under ROC curve) of 0.69 (95% confidence interval between 0.60 and 0.79; Fig. 3a and Supplementary Data 4). We confirmed that this performance is robust with regard to the choice of a wide range of model parameters (Supplementary Fig. 2), but required all features combined, as classification using

individual features is less accurate (AUC < 0.64; Supplementary Fig. 3). Classification using several other machine learning methods, namely logistic regression, support vector machine (SVM), XGBoost and neural network, also resulted in similar performance (ROC AUC: 0.69-0.73; Supplementary Fig. 4a-d). Furthermore, including nucleotide and overlapping di-nucleotide identities further increased ROC AUC to 0.74 (95% confidence interval between 0.65 and 0.83; Supplementary Fig. 5a), consistent with the difference between crosslinked and non-crosslinked nucleotides in base composition, as we identified above.

Random forest models provide ranks of features based on their importance for classification reflected in a reduction in Gini index[38]. Since our sample size used for classification is relatively small, we took caution to ensure the ranks of features are robust using permutation tests. In addition, we also used a generalized linear regression model to determine whether each feature contributes positively or negatively to crosslinking, which was not provided by the random forest model (see Methods). Consistent with results obtained from analysis of individual features, we found that planar base stacking with phenylalanine, or hydrophobic amino acids as a group, and the C2′-endo conformation of sugar puckering represent the top-ranked features that facilitate crosslinking (Fig. 3b, Supplementary Fig. 5b and Supplementary Data 5). Interestingly, there is also an indication that certain types of hydrogen bonds, such as those formed between the phosphate (PO4) group and the sidechain of arginine, may contribute to crosslinking.

Together, our random forest prediction and analysis of individual features suggest that nucleotide base stacking with aromatic residues represents a prominent structural feature that facilitates crosslinking. However, we also noticed that only 22 of 76 (29%) stacking interactions had detected crosslinking, while the remaining 54 (71%) cases did not. To investigate additional structural features that may contribute to crosslinking cooperatively with base-stacking, we focused on the 76 nucleotides stacking with aromatic residues and built another random forest prediction model. In this analysis, we achieved a prediction accuracy ROC AUC of 0.80 (95% confidence interval between 0.68 and

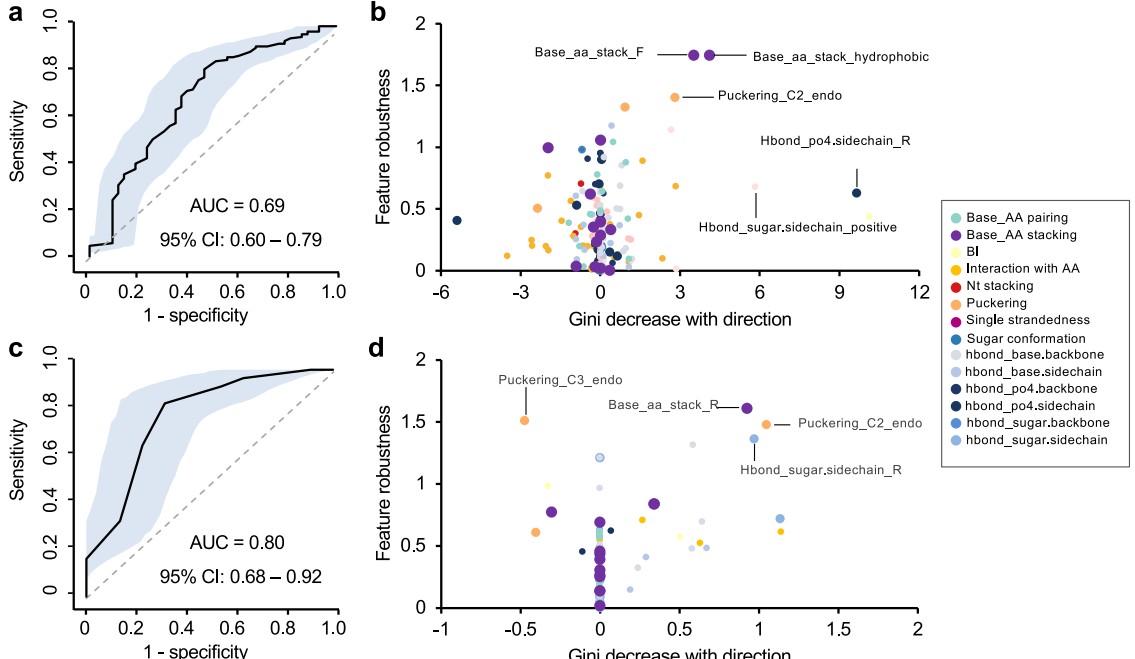

**Fig. 3 | Prediction of crosslinked vs. non-crosslinked nucleotides using random forest. a** Prediction performance of crosslinked vs. non-crosslinked nucleotides as measured by AUC (black curve). The shaded area indicates 95% confidence interval as determined by 2000 models trained with bootstrapped data. **b** Feature importance plot with Mean GiniDecrease of each feature shown in x-axis and feature robustness derived from permutation tests shown in y-axis. The direction of Mean GiniDecrease represents whether the feature is positively or negatively associated with crosslinking. Different feature groups are color-coded. **c**, **d**, Similar to (**a**, **b**) but the analysis is limited to crosslinked vs. non-crosslinked nucleotides stacking with aromatic amino acids.

0.92; Fig. 3c and Supplementary Fig. 5c). Evaluation of feature importance for prediction suggests that the C2′-endo sugar puckering type, as well as interaction with arginine through hydrogen bond and base stacking, contributes to protein–RNA crosslinking (Fig. 3d and Supplementary Fig. 5d). The presence of such structural features appears to be particularly common for RRM, as shown in the example of RBFOX. In this case, G6 of the UGCAUG motif interacts with phenylalanine (F160) and arginine (R118) through base stacking and hydrogen bond, respectively, and both amino acids were found to be crosslinked (Fig. 1b). Taken together, our analysis of structural features associated with crosslinked nucleotides identifies the structural requirements to induce protein–RNA photo-crosslinking and highlights the importance of base stacking with phenylalanine, specific types of hydrogen bonds and nucleotide conformation in this process.

## Distinct structural features associated with crosslinked amino acids

To further validate our finding regarding the structural features associated with crosslinked nucleotides, we applied the PxR3D-map method to analyze crosslinked amino acids identified by RNA-interactome capture. Among the previous studies that reported crosslinked amino acids at single residue resolution[20–22], we decided to focus our analysis on data obtained from RBS-ID, which represents the most comprehensive collection of crosslinked amino acids[21]. After intersecting with protein–RNA complex structures in PDB, we obtained 55 nonredundant complexes (Supplementary Data 6), which consisted of 116 crosslinked amino acids and 1,380 non-crosslinked amino acids that are in direct contact with RNA. For each of these amino acids, we extracted 36 structural features describing the identity of the interacting nucleotides, and the type of protein–RNA contacts including hydrogen bonds, planar side-chain base stacking, and pseudo pairing (Supplementary Datas 2 and 7).

Among all crosslink sites identified by RBS-ID without filtering by protein–RNA complex structures, the most abundant amino acids

reported in the original study were cysteine, followed by phenylalanine, tyrosine, and arginine[21], which was reproduced in our analysis (Fig. 4a). The enrichment of phenylalanine, tyrosine, and arginine is consistent with our analysis of amino acids interacting with crosslinked nucleotides, as described above. On the other hand, while cysteine was also known to be susceptible to UV crosslinking with nucleotides[15,39], it was not enriched among amino acids directly contacting crosslinked nucleotides. Interestingly, while 67% of crosslinked phenylalanine and 43% of crosslinked tyrosine are located within annotated RBDs, the proportion is much lower for cysteine (29%) (Fig. 4a).

The under-representation of crosslinked cysteine within annotated RBDs, as compared with the other most frequently crosslinked amino acids, aromatic residues in particular, is intriguing. We hypothesize that this could be potentially explained by two mechanisms. One hypothesis is that due to its high photoreactivity, cysteine can get crosslinked with nucleotide even when the protein–RNA interaction is transient. Alternatively, RBPs without annotated RBDs (unconventional RBPs or ucRBPs[40]) may have distinct modes of protein–RNA interactions with comparable structural stability as complexes formed between RNA and conventional RBPs (cRBPs) with well-characterized RBDs. Cysteine can be involved in these unconventional protein–RNA interactions, resulting in its increased crosslinking frequency. We prefer the first hypothesis because we and others recently found that unconventional RBPs rarely have RNA sequence specificities[40,41].

To formally distinguish these two hypotheses, we first examined the crosslinked amino acids located in well-characterized RBDs with experimentally resolved protein–RNA structures. We asked if the crosslinked amino acids differ depending on whether they are at the protein–RNA interaction interfaces, with the assumption that amino acids at the interaction interfaces (i.e., RNA-contacting amino acids) most likely participate in stable interactions with RNA, while those amino acids outside the interaction interfaces (non-contacting amino acids) more likely involve transient contacts with RNA. We found that the proportion of crosslinked amino acids directly contacting RNA is

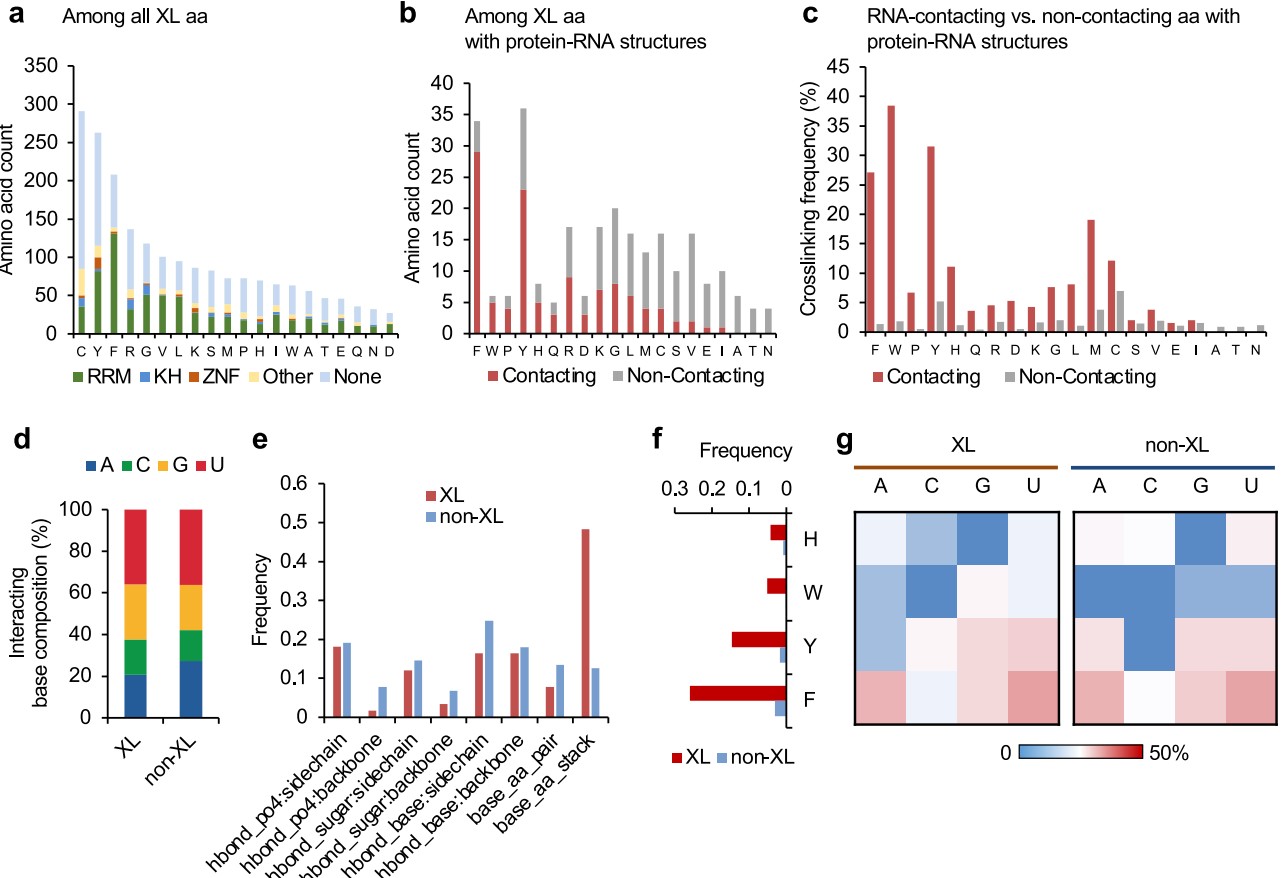

**Fig. 4 | Comparison of crosslinked and non-crosslinked amino acids using associated structural features. a** The number of crosslink sites grouped by amino acid and RBD types. Amino acids are ordered by the total number of crosslinked sites. **b** Among the crosslinked amino acids that can be mapped to structurally resolved protein–RNA complexes, the number of crosslink sites with and without direct RNA contacts is shown. Amino acids are ranked by the proportion of RNA-contacting crosslink sites. **c** Amino acids in structurally resolved protein–RNA complexes are divided into two groups based on whether they directly contact RNA at the protein–RNA interaction interface, and the crosslinking frequency is calculated for each amino acid and group separately. Amino acids are ranked as in panel (**b**). **d** Among all RNA-contacting amino acids in structurally resolved protein–RNA complexes, the base composition of the closest nucleotides is shown for crosslinked and non-crosslinked sites separately. **e** Frequency of structural features associated with crosslinked and non-crosslinked amino acids. **f** Frequency of base stacking interactions for aromatic residues at crosslinked and non-crosslinked sites. **g** Similar to (**f**) but shown for individual nucleotide bases separately in heatmaps.

also much lower for cysteine (4/16 = 25%), as compared to phenylalanine (29/34 = 85%) and tyrosine (5/6 = 83%) ($p = 3.5e{-}5$, Fisher's exact test) (Fig. 4b). The difference is not due to a general depletion of cysteine at the protein–RNA interaction interfaces, as compared to the other most frequently crosslinked amino acids. Indeed, when we estimated the crosslinking frequencies for RNA-contacting vs. non-contacting amino acids separately, cysteine is the most frequently crosslinked amino acid in the non-contacting group (7%), followed by methionine (3.8%), valine (1.9%), and tryptophan (1.85%); non-RNA-contacting phenylalanine has a crosslinking frequency of 1.3%. A distinct crosslinking profile was observed for the RNA-contacting amino acid group, with dramatically increased crosslinking frequencies for all aromatic residues (phenylalanine: 27% or 20 fold, tryptophan: 38.5% or 21 fold, tyrosine: 31.5% or 6.1 fold, histidine: 11% or 9.1 fold) and methionine (19% or 5.0 fold), while cysteine has only a moderate increase in crosslinking frequency (12% or 1.7 fold) (Fig. 4c). We also directly compared amino acid crosslinking frequencies between conventional and unconventional RBPs, using the list of proteins compiled by Ray et al. [40]. Compared to conventional RBPs, unconventional RBPs are generally enriched in cysteine (17.2% vs. 7.5%) and depleted in phenylalanine (7.2% vs. 15.1%; Supplementary Fig. 6a). Importantly, they also have a higher crosslinking frequency of cysteine (11.3% vs. 7.6%) but a lower crosslinking frequency of phenylalanine (2.2% vs. 5.8%; Supplementary Fig. 6b), which is qualitatively similar to the

differences we observed between RNA-contacting and non-contacting amino acids in conventional RBPs with defined structures. Together, these data imply that cysteines can be crosslinked regardless of whether they are at the protein–RNA interaction interfaces of stable complexes.

We also examined the base composition of the closest nucleotides each crosslinked vs. non-crosslinked amino acid directly contacts, as a proxy of the nucleotide it might crosslink to. In this analysis, the base compositions for the two groups are in general similar (p = 0.25, chi-squared test), with only a slight enrichment of G/U for crosslinked amino acids (Fig. 4d). Finally, we examined the types of amino acid-nucleotide contacts, and found that the only type of contacts enriched in crosslinked vs. non-crosslinked amino acids is base stacking (0.53 vs 0.13 event per amino acid; Fig. 4e). This enrichment was observed for all four aromatic residues (Fig. 4f) and they do not appear to show an overt bias for particular nucleotides, as compared to non-crosslinked amino acids interacting with RNA (Fig. 4g). This analysis confirmed that base stacking between aromatic residues and RNA nucleotides can strongly facilitate photo-crosslinking.

## Prediction of crosslinked amino acids based on structural features

We next focused on the 116 crosslinked and 1,380 non-crosslinked amino acids that are in direct contact with RNA and applied PxR3D-

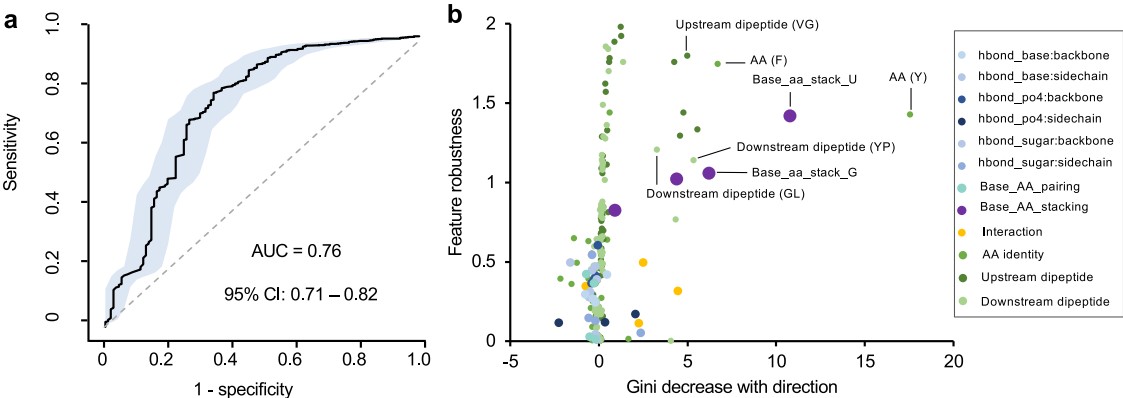

**Fig. 5 | Prediction of crosslinked vs. non-crosslinked amino acids using random forest. a** Prediction performance of crosslinked vs. non-crosslinked amino acids as measured by AUC (black curve). The shaded area indicates the 95% confidence interval as determined by 2000 models trained with bootstrapped samples. **b** Feature importance plot with Mean GiniDecrease of each feature shown in the x-axis and feature robustness derived from permutation tests shown in the y-axis. The direction of Mean GiniDecrease represents whether the feature is positively or negatively associated with crosslinking. Different feature groups are color-coded.

map to predict crosslinked vs. non-crosslinked amino acids using structural features. A random forest model was trained using the amino acid and dipeptide identities as well as the 36 structural features describing amino acid-nucleotide contacts. In this analysis, we achieved an ROC AUC of 0.76 by 10-fold cross-validation (95% confidence interval between 0.71 and 0.82; Fig. 5a and Supplementary Data 7). Again, the performance of the random forest model is robust with respect to a wide range of model parameter choices (Supplementary Fig. 7), but classification using individual features is less accurate (AUC < 0.59; Supplementary Fig. 8). Classification using other machine learning methods also resulted in very similar performance (Supplementary Fig. 9a-d). By applying the same feature ranking method as described above, we found aromatic residues including phenylalanine and tyrosine as well as stacking of these residues with nucleotide bases, especially uridine and guanine, as the most strongly associated features for prediction. Interestingly, we found that several dipeptides involving glycine, such as VG and GL, are also ranked among the top features (Fig. 5b; also see below). Consistent with our previous analysis of crosslinked nucleotides, most types of hydrogen bonds seem to have no or only moderate contribution to facilitating crosslinking (Fig. 5b and Supplementary Data 8).

### Different mechanisms of photo-crosslinking for RRMs and KH domains

Taking advantage of the expanded number of crosslink sites identified by RNA-interactome captures, we next examined whether different types of RBDs have different crosslinking mechanisms. It is well known that different RBDs have distinct amino acid compositions and structural characteristics at the protein–RNA interaction interface. For example, RRMs and K homology (KH) domains are the two most abundant types of RBDs with distinct folds and RNA binding specificities. While aromatic residues, which we found to facilitate crosslinking, are frequently found in RRMs, they are rarely present in KH domains[42,43].

By comparing the residue composition of crosslinked amino acids in different types of RBDs, we found aromatic residues, such as phenylalanine and tyrosine, are indeed the most abundant at the crosslink sites in RRMs. This is in stark contrast to KH domains, in which crosslinking frequently occurs at cysteine, arginine, and glycine. For zinc finger (ZNF) domains, tyrosine is the most represented amino acid (Fig. 6a).

To investigate unique mechanisms of protein–RNA crosslinking that may underlie specific types of RBDs, we focused on the comparison of RRMs and KH domains. For each type, we compiled a list of domains, including those with identified crosslink sites. Multiple

sequence alignments were then performed, so that the crosslink sites in different domains of each type could be compared (Supplementary Figs. 10 and 11). This analysis confirmed that the major crosslink sites for RRMs are located in the two ribonucleoprotein domains RNP-1 and RNP-2, which are two critical features for RNA binding in RRMs. There, the two conserved phenylalanines are the most frequent crosslink sites (Fig. 6b), and they typically form base stacking with RNA (Fig. 1b and additional examples discussed in refs. 20,21,23). In contrast, for KH domains, the most frequent crosslink sites are located adjacent to the GXXG (X=any amino acid) motif, a defining feature of the KH domain, in the N-terminus; these crosslink sites frequently involve a dipeptide including glycine, such as cysteine-glycine (Fig. 6c). In general, cysteine is crosslinked in the context of the cysteine-glycine dipeptide, while glycine is crosslinked in GX or XG dipeptide in the absence of cysteine. However, crosslinking rarely occurs in the GXXG motif.

KH domain is known to recognize an RNA tetramer motif, such as YCAY (Y = C/U) for NOVA[43–46]. When we examined these protein–RNA complex structures in detail, we found that the dipeptide at the crosslink site interacts with the first position of the tetramer. For example, glycine (G100) has evidence of crosslinking in QKI STAR domains (a variant of KH domains with two flanking Qua domains)[21], and it interacts with the first uridine of the UAAC motif for QKI (Fig. 6d; in this case, the first two nucleotides in the QKI binding sequence motif ACUAAC are recognized by the Qua2 domain)[47]. In the case of NOVA, the crosslink site in the protein is unknown. However, we previously determined that the first uridine in the YCAY motif represents the predominant crosslink site in RNA. This nucleotide contacts a glycine-alanine dipeptide in NOVA2 KH3 (G18-A19, PDB accession: 1EC6; ref. 45; Fig. 6e). Importantly, in both cases, the GX or XG dipeptide bond appears to stack over the base of the contacting nucleotide[48]. This is confirmed by a systematic search for planar stacking formed between dipeptide bonds and nucleotide bases using 3DNA-SNAP[27,28]. Furthermore, our survey also found additional examples including NOVA1 KH3 (G18-A19; PDB accession: 2ANN[48]), PCBP2 KH1 (G26-S27; PDB accession: 2PY9, ref. 49.) and KH3 (G300-S301, PDB accession: 2P2R[50]). This analysis suggests that crosslinking of KH domains with RNA utilizes a distinct mechanism from base stacking over aromatic residues observed for RRMs.

### Protein–RNA crosslinking in the ribosome

Our analyses so far have focused on relatively simple complexes formed between monomer RBP and RNA ligand. To test whether conclusions from these analyses are generalizable to large protein–RNA complexes which involve more non-sequence specific and dynamic interactions, we examined the ribosome using an 80S

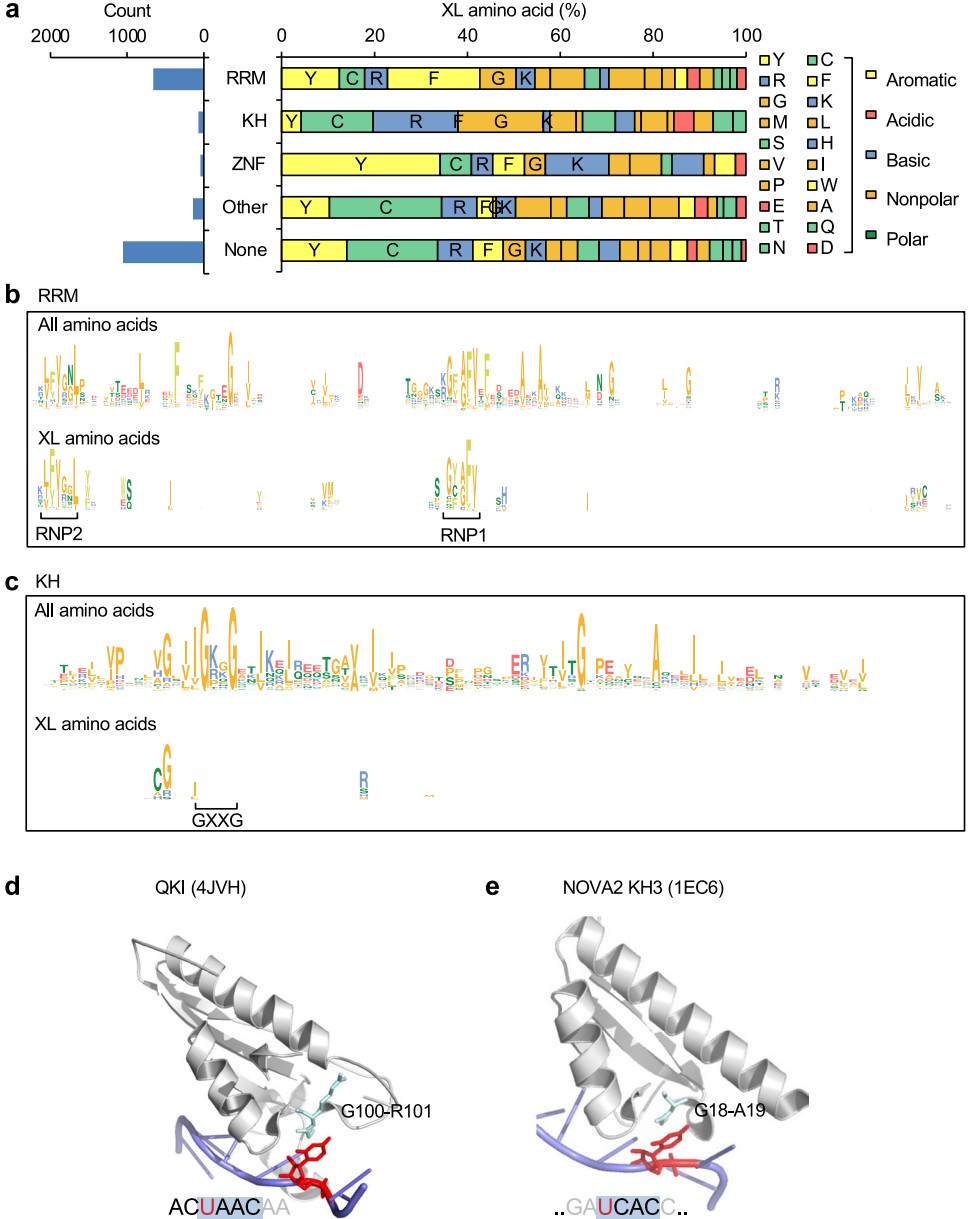

**Fig. 6 | Distinct protein–RNA crosslinking mechanisms for different types of RBDs. a** The number (left) and composition (right) of crosslinked amino acids in different types of RBDs. **b** Top: Multiple sequence alignment logo for RRMs. The total height of each position shows the information content of all amino acids in each position, while the relative height of each letter reflects the amino acid composition at that position. Amino acid color codes are the same as in (**a**). Bottom: Crosslinked amino acids at each position are shown in the same format as at the top. Note that crosslinked amino acids are visible only for positions with non-zero information content. The two RNP motifs defining RRMs are indicated. **c** Similar to (**b**) but for KH-domains. The GXXG motif defining the KH-domains is

indicated. **d**, **e** Examples of KH-domains in which a glycine-related dipeptide stacks over the base of the first position of its tetramer RNA sequence motif. In each panel, the structure is illustrated using PyMOL[67], with protein in gray and RNA in blue cartoons. The RBD and PDB accession codes are indicated. **d** The GX dipeptide shown in cyan (stick) indicates crosslinked amino acids. The dipeptide bond stacks over the first position (red) of the tetramer RNA motif, highlighted using a shaded box in the RNA nucleotide sequence. **e** The first nucleotide (red) of the UGAC motif represents the predominant crosslink position determined using CLIP data. The GA dipeptide interacting with this uridine is highlighted in cyan (stick).

ribosome structure determined by Cryo-EM[51]. This structure contains 79 proteins, from which we identified 272 crosslinked amino acids using RBS-ID data (Supplementary Data 9). Among them, phenylalanine, tyrosine, arginine, lysine and cysteine are the most crosslinked amino acids (Fig. 7a). We then analyzed crosslinked amino acids separately depending on whether they directly contact RNA in the structure. When normalized by amino acid compositions in ribosomal proteins in the structure, cysteine showed the highest crosslinking efficiency (12.8%) among amino acids without direct contact with RNA,

followed by aromatic residues phenylalanine (3.3%), tyrosine (4.8%) and tryptophan (6.1%). Furthermore, in comparison between RNA-contacting vs. non-contacting amino acids at crosslink sites, aromatic residues phenylalanine, tryptophan and tyrosine showed a larger increase in crosslinking efficiency, as compared to cysteine (2.9-3.6 fold vs. 2.3 fold) (Fig. 7b). Among these three aromatic residues interacting with RNA through base stacking, 29.5% showed evidence of crosslinking (See examples in Supplementary Fig. 12). These data confirmed that cysteine is the most photoreactive amino acid even

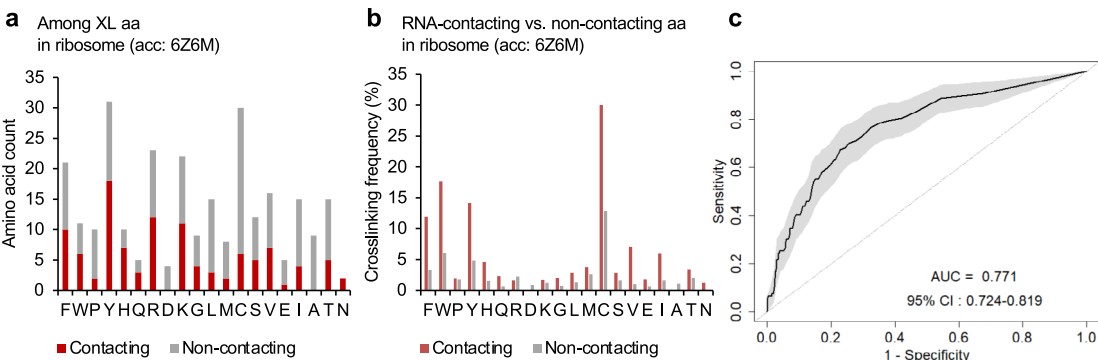

**Fig. 7 | Comparison of crosslinked and non-crosslinked amino acids in 80S ribosome using associated structural features. a** Among the crosslinked amino acids that can be mapped to the ribosome structure (PDB accession: 6Z6M), the number of crosslink sites with and without direct RNA contacts is shown. Amino acids are ranked as in Fig. 4b. **b** Amino acids in the ribosome structure are divided into two groups based on whether they directly contact RNA at the protein–RNA interaction interface, and the crosslinking frequency is calculated for each amino acid and group separately. Amino acids are ranked as in Fig. 4b. **c** Prediction performance of crosslinked vs. non-crosslinked amino acids as measured by AUC (black curve). The shaded area indicates the 95% confidence interval as determined by 2000 models trained with bootstrapped samples.

when the residue does not stably interact with RNA, while aromatic residues facilitate protein–RNA crosslinking through base stacking. Applying the same random forest models for the classification of crosslinked vs. non-crosslinked amino acids resulted in an ROC AUC of 0.77, which is also consistent with our results from individual protein–RNA complexes (Fig. 7c).

## Discussion

In this study, we developed a computational framework and method named PxR3D-map to systematically analyze structural features associated with and likely contributing to photo-crosslinking between protein and RNA in their native complexes. Given the wide applications of UV-crosslinking to help interrogate protein–RNA interactions and their functional consequences in gene regulation, the identification and characterization of the precise crosslink sites in both RNA and protein is undoubtedly a key step to better interpret data generated by these assays and understand mechanisms underlying protein–RNA complex formation.

Although current technologies fall short in the simultaneous identification of crosslink sites in RNA and protein, we demonstrate that this limitation can be mitigated by the integration of crosslink sites identified separately in protein or RNA for protein–RNA complexes with experimentally resolved 3D structures. Systematic analysis of an expanding list of complexes allowed us to gain mechanistic insights into protein–RNA crosslinking. Based on our analysis, direct contact between amino acids and nucleotides is clearly not sufficient to induce the formation of covalent bonds between the two in the experimental conditions currently used to study protein–RNA complexes in vivo or in cells. Instead, this process relies on certain structural features in the complex. Even though currently resolved protein–RNA complex structures in the PDB are still limited, several structural features facilitating protein–RNA crosslinking start to emerge. Among them, the contribution of base stacking with peptide side chains, especially aromatic residues, is the most compelling. Crosslinking at such interaction sites has been experimentally validated between LIN28 CSD and its binding motif UGAU. In this case, the last uridine stacks with a phenylalanine and was crosslinked when the complex reconstituted in vitro was irradiated by UV light[24].

While this manuscript was in preparation, a study was published that characterized the crosslinking of RBFOX1 RRM with the cognate binding sequence UGCAUGU in detail[23]. The authors combined mass-spectrometry analysis with isotope labeling of specific nucleotides to determine crosslink sites in RNA and protein simultaneously. This study validated our previous finding from CIMS/CITS analysis that the

two guanines in the motif are predominant crosslink sites in RNA[18], while also determining the two phenylalanine residues that stack with the two guanines as crosslink sites in the protein. By manual examination of aromatic residues and base stacking around crosslink sites determined by RNA-interactome capture, they independently identified the importance of stacking between the aromatic ring and the nucleotide base for photo-crosslinking. While the exact mechanisms of peptide-nucleotide adduct formation through base stacking are yet to be established, possible mechanisms have been proposed[23]. We note that it has been known for decades that UV light can induce the formation of U·U or U·C dimers for nucleotide bases stacked together[52], probably through similar mechanisms.

Our systematic and unbiased study, which is made possible by automated analysis of protein–RNA complex structures and structural feature extraction followed by rigorous machine learning-based classification, allowed us to identify additional structural features associated with crosslink sites. Among them, the most notable finding is the crosslinking of dipeptides is most likely through nucleotide base stacking over the dipeptide bond. A systematic search for such dipeptide bonds revealed that they are most frequently associated with glycine, probably because of the small size and structural flexibility of this amino acid. This mechanism appears to be particularly important for crosslinking of KH domains with RNA. In addition, we also noticed that several other residues including valine and lysine are frequently crosslinked in RNP-1 and RNP-2 of RRMs. Moreover, certain types of hydrogen bonds, as well as the C2′-endo type of nucleotide conformation, may also contribute to crosslinking. This observation is consistent with previous work that demonstrated that DNA in the C2′-endo conformation is less UV-resistant compared to the C3′-endo conformation[53]. Altogether, we propose that multiple mechanisms underly the selective photo-crosslinking between protein and RNA, which await further validation and characterization. It is also important to note that our method aims to reveal common structural features shared across many complexes, while protein-specific or rare features contributing to the photocrosslinking efficiency will be missed and they have to be investigated using alternative methods interrogating individual protein–RNA complexes (e.g., refs. 23,24).

Knorlein et al. reported that up to 78% of crosslink sites in structurally resolved protein–RNA complexes can be explained by base stacking with aromatic residues[23]. This estimate appears to be rather high based on our analysis, even when we focused on RRMs, in which aromatic residues are overrepresented at the protein–RNA interaction interface (Fig. 6b). Moreover, crosslinking through base stacking with aromatic residues is clearly not the main mechanism for KH domains,

which are the second most common type of RBDs (Fig. 6c). Along this line, our analysis suggests that stacking between RNA base and aromatic residues is insufficient to induce crosslinking. A striking example illustrating this point is PUM, which recognizes each of the eight nucleotides in its RNA sequence motif through base stacking, and yet crosslinking occurs specifically at the first uridine of the motif or one nucleotide further upstream (Fig. 1d). This is also in line with our finding that additional structural features, including hydrogen bonds between the sugar of the nucleotide and the side chain of arginine, as well as C2′-endo sugar puckering, can classify crosslinked vs. non-crosslinked stacking interactions (Fig. 3c, d).

Our analysis provides insights into the crosslinking of cysteine, which is the most photoreactive among amino acids due to the presence of sulfur as an electron donor[15,39]. While cysteine is the most abundant amino acid at crosslink sites identified by RNA-interactome capture[21], a large proportion of them are not located in known RBPs. Even for those located in the annotated RBDs of RBPs, they frequently do not directly contact RNA, as judged from available protein–RNA complex structures (Fig. 4a, b). This is in contrast to the crosslinking of aromatic residues, which is found predominantly at the protein–RNA interaction interfaces (where they form base stacking with nucleotides). In addition, for unconventional RBPs identified by RNA interactome captures, the crosslinking of cysteine is much more frequent while the crosslinking of aromatic residues is much less, as compared to conventional RBPs. These observations are most consistent with the notion that crosslinking of cysteine with RNA can occur independently of stable protein–RNA interactions. That said, we note that among the 13 crosslink sites found in KH domains that involve cysteine, 10 (77%) occur in the CG or GC dipeptide context. This proportion does not seem to be explained by the overrepresentation of such dipeptides in KH domains in general and may reflect the importance of these dipeptides in initiating crosslinking at cysteine.

Crosslinking bias in CLIP data has been discussed in the literature[54], but the extent and underlying mechanisms are not clear. Following our analyses, such bias can arise when the structural requirement to induce crosslinking cannot be fulfilled in protein–RNA contacts that determine specific protein–RNA interactions, but is rather fulfilled through additional protein–RNA contacts which might be more transient. We previously found cases in which the crosslink sites are not part of the RBP binding motifs but are located in the immediate vicinity, such as crosslinking of the upstream uridine in PUM binding motif (Fig. 1d). As another example, SRSF1 binds to GGAGGA or the half site GGA, but crosslinking occurs predominantly in an upstream uridine in the sequence UGGA and the uridine does not contribute to binding specificity or affinity[41].

Indeed, crosslinking of transient protein–RNA interactions without stable complex formation could be widespread. This is in line with the overrepresentation of cysteine among crosslinked amino acids with a majority lacking evidence of direct protein–RNA contacts from stable complexes, as described above. In addition, we noticed that among crosslink sites in RNA identified by CIMS and CITS analysis of RBFOX CLIP data[18], over half are uridine when the consensus UGCAUG or UGCAUG-like (with one mismatch) sequences are not present. On the surface, this observation appears to be inconsistent with the predominant crosslinking of RBFOX with guanines when stable complexes are formed between the RRM and UGCAUG element. However, this apparent discrepancy can be resolved if one envisions that crosslinking could occur when the protein scans through RNA before a cognate binding site is found. In this context, multiple interactome capture studies aimed to identify the crosslinked peptide-nucleotide adducts reported that the crosslinked RNA moiety is predominantly uridine[20,21,25] and Bae et al. decided to focus on uridine exclusively in their mass-spectrometry analysis to limit the search space[21]. Our analysis implies that a subset of the crosslink sites with uridine adducts

could reflect transient protein–RNA interactions which might outnumber stable complexes. From these experiences, the identification of predominant crosslink sites (especially uridine) outside the expected sequence motif or lack of specific sequence motifs around crosslink sites beyond uridine-rich sequences should signal caution in CLIP data analysis. Importantly, the interpretation of protein–RNA interactions mapped by CLIP can benefit from validation using independent approaches such as assessment of binding specificity by in vitro binding assays without UV crosslinking[55,56], experimental mutagenesis, or allele-specific binding analyses using CLIP data[41]. In this regard, it is worth noting that a high-throughput method was developed recently to map RBP binding sites by in situ reverse transcription and sequencing without relying on UV-crosslinking[57].

We estimate that about 55–60% crosslinked amino acids identified by RNA-interactome captures are not located in the stable protein–RNA interaction interface of individual protein–RNA complexes or the ribosome. Together with the overrepresentation of cysteine crosslinking, this study also warrants caution for the numerous RBPs identified by RNA-interactome captures, including many unconventional RBPs without any characterized RBDs. Along this line, a recent study proposed that the larger set of RBPs identified by interactome capture should not be conflated with the conventional subset because the vast majority of unconventional RBPs lack any apparent sequence specificity, similar to an observation we made previously based on CLIP data analysis[40,41]. The proposal is echoed and extended in this study, as we now provide a mechanistic explanation— they tend to be pulled down due to protein–RNA crosslinking during transient contacts. That said, one should certainly not exclude the possibility that a subset of unconventional RBPs might have the capacity to form stable protein–RNA complexes similar to their conventional counterparts, or that transient protein–RNA interactions might have certain functional significance. We examined the list of unconventional RBPs previously shown to have clear sequence specificity and found two RBPs with high-resolution protein–RNA complex structures and evidence of crosslinking. Among them, ZRANB2 contains two zinc fingers, with each zinc finger recognizing a 5′ splice-site-like AGGUAA motif based on in vitro binding[40,58] and CLIP data. Our CLIP data analysis also determined that the predominant crosslinking occurs at G2 of this motif, which forms base stacking with an aromatic residue tryptophan (W79; PDB accession: 3G9Y) (Supplementary Fig. 13a). In the other example, RPP25 crosslinks with RNA through a cysteine (C70; PDB accession: 6LT7). However, this amino acid does not directly interact with RNA (Supplementary Fig. 13b).

Finally, the specific structural requirement for protein–RNA crosslinking suggests that the crosslink sites mapped by CLIP or RNA-interactome capture could provide useful constraints when one develops models of protein–RNA complex structures. This direction is particularly encouraging because of recent progress made in the computational prediction of protein structures using primary sequences[59]. These models are typically trained with large datasets, a requirement that is difficult to meet for protein–RNA complexes (~2000 structures in total as of June 2022). It is possible that the demands on a large number of training complexes could be relaxed by providing structural constraints for protein–RNA complexes as suggested by selective protein–RNA crosslink sites. On the other hand, the predicted protein–RNA complex structures may also facilitate the characterization of protein–RNA crosslinking by expanding the sample size that can be used by PxR3D-map analysis, a major limitation of the current study. Moreover, we expect that experimental mapping of protein–RNA interactions can benefit from technologies that are able to crosslink protein and RNA with less selective structural requirements to achieve improved sensitivity and reduced bias. The methodological framework and analyses presented in this work could provide a guide for the future development of such technologies.

## Methods

### Compilation of protein–RNA complex structures and structural feature extraction

For systematic analyses of protein--RNA complex structural features described in this study, we downloaded and parsed all Protein Data Bank (PDB)[26] structures to identify macromolecular complexes involving both protein and RNA chains in July 2020. In total we identified 1,090 complexes, including 277 (human), 28 (mouse), and 6 (rat) complexes, respectively.

All these complexes were analyzed by the programs DSSR and SNAP in the 3DNA software suite[27,28] to determine nucleotides and amino acids in direct contacts (defined by the shortest distance between a pair of heavy atoms in the nucleotide and the amino acid ≤4.5 Å). In addition, DSSR and SNAP extracted structural features that describe RNA nucleotide conformation (e.g., base morphology and sugar puckering), RNA-secondary structures (e.g., single vs. double-stranded region), and various types of RNA-protein contacts including different types of hydrogen bonds, planar amino acid sidechain-base stacking, and pseudo pairing. Occasionally, amino acid-RNA contacts of particular types have distances slightly above the threshold and these were also included. To facilitate downstream analysis, we summarized these features with respect to their association with each nucleotide in the RNA ligand (e.g., the number of hydrogen bonds with each of the 20 amino acids; 246 features in total; Supplementary Data 1) or each residue in the protein chain (e.g., the number of hydrogen bonds with each of the four RNA bases; 36 features in total; Supplementary Data 2) at the protein–RNA interaction interface. Additional features were also included by aggregating amino acids with similar properties (6 categories: polar, positive, negative, hydrophobic, aromatic and aliphatic; Supplementary Data 4).

### Mapping crosslink sites to structurally resolved protein–RNA complexes

To map crosslinked nucleotides of RNA ligands in structurally resolved protein–RNA complexes, we intersected the RBPs in these complexes and those with CLIP data and obtained a list of 41 RBPs[18,60–64]. Crosslinking sites in RNA were mapped by crosslinking-induced mutation site (CIMS) and truncation site (CITS) analysis in our previous studies (e.g., ref. [41]) or performed in this study using the CLIP Tool Kit (CTK) package[19]. De novo motif analysis was performed using mCross[41]. For this study, we kept only RBPs whose consensus binding motif can be unambiguously determined and matches the RNA ligand used in structure determination. For each RNA ligand, we then searched instances of the RNA sequence in the respective CLIP data to determine the frequency of crosslinking at each nucleotide position using the mCross package[41]. When the RNA ligand is long and only part of the sequence interacts with RBP, the sub-sequence that resembles the consensus RNA-binding sequence motif of the RBP was used for the search. In these cases, searches with one or two nucleotide extension on each side were also performed to identify additional crosslink sites immediately flanking the consensus motif sequence. A nucleotide in the RNA ligand was defined as crosslinked nucleotide if we found ≥20 instances of the ligand (or its subsequence interacting with the protein) with crosslink sites determined by CLIP, and the crosslinking frequency at the position among all instances was ≥0.3. Alternatively, when a smaller number of instances with crosslinking evidence were available (≥10), we required the crosslinking frequency at the position ≥0.5. The remaining positions of the RNA ligand interacting with the protein were defined as non-crosslinked nucleotides (Supplementary Data 4). Positions without direct contact with amino acids were excluded in our analysis. When multiple protein–RNA complex structures were available for the same RBD, only the one supported by the most crosslinking events was kept in our analysis. We were able to determine crosslink sites in RNA unambiguously for 29 non-redundant protein–RNA complexes representing 25 RBPs, from which we obtained 43 crosslinked and 171 non-crosslinked nucleotides directly contacting the protein (Supplementary Datas 3 and 4).

To map crosslinked amino acids in structurally resolved protein–RNA complexes, we used crosslink sites identified by RBS-ID[21]. This dataset included 1,970 crosslink sites in proteins (denoted RNA-binding sites or RBSs in the original paper) that belong to 640 protein groups. The amino acid coordinates reported for these sites were compared with the amino acid residue coordinates in the protein chain in the respective PDB structures. In total, we found 104 complex structures from 46 proteins with at least one crosslinked amino acid. We removed the redundant structures and chains by keeping the one with the maximum crosslinked peptide counts and the maximum number of non-zero structural features to obtain 55 non-redundant complexes (Supplementary Data 6). After removing amino acids without directly contacting RNA nucleotides, we obtained 116 crosslinked and 1,380 non-crosslinked amino acids used in our analysis (Supplementary Data 7).

We also examined the crosslinked amino acid compositions in 398 conventional RBPs (cRBPs) with annotated RBDs and 624 unconventional RBPs (ucRBPs) lacking annotated RBDs, as defined by Ray et al. [40]. Among 1,970 crosslink sites identified by RBS-ID, 1,029 and 458 crosslinked amino acids were mapped to 152 cRBPs and 167 ucRBPs, respectively. Finally, we analyzed 80S ribosome structure (PDB accession: 6Z6M[51]), which includes 79 proteins. In total, RBS-ID detected 288 crosslinked amino acids, among which 272 residues can be mapped to the PDB structure (107 RNA-contacting and 165 non-contacting residues, respectively; Supplementary Data 9). Structural features associated with each amino acid were extracted as described above.

### Prediction of crosslinked nucleotides and amino acids using random forest models

We trained a random forest model to predict crosslinked vs. non-crosslinked nucleotides using structural features as described above and prioritize the structural features that contribute to protein–RNA crosslinking. Another random forest model was developed to predict crosslinked vs. non-crosslinked amino acids and prioritize associated structural features. For these tasks, the Caret package (version 6.0.80; ref. [38]) in R was used to train the models and perform predictions.

We observed improved sensitivity without impairing specificity by adopting a sampling method that corrects the imbalance of the positive and negative samples using SMOTE[65]. Model performance was evaluated by 10-fold cross-validation using ROC area under curve (AUC). To evaluate the robustness of the models with respect to model parameter choices, we trained models with different parameters including the number of trees (ntree) in the forest and the number of features per tree (mtry). For the prediction of crosslinked nucleotides, the optimal performance was achieved with mtry = 17 and ntree = 100 as measured by AUC (Fig. 3a). Similarly, for the prediction of crosslinked amino acids, the optimal performance was achieved with mtry = 6 and ntree=1000 for individual RBP-RNA complexes (Fig. 5a) and mtry = 6 and ntree=100 for ribosomal proteins (Fig. 7c).

### Evaluation of structural feature importance

For a trained random forest model, Mean GiniDecrease was used to rank the feature importance for prediction. Given the relatively moderate sample size of data available for our classification tasks, we also developed a metric to measure the robustness of feature ranks. Specifically, we randomly permutated the crosslinking labels of the nucleotides (or amino acids) and re-trained the models with the same parameters. For each permutation and re-trained model, the features were ranked. This process was repeated for $N = 2000$ times, and the robustness of each feature is defined as $R_s = -\log_{10}(\sum_{i=1}^{N} I(r_{is} < r_s)/N)$, where $r_s$ is the rank of feature importance for feature $s$ in the true

model without crosslinking label permutation and $r_{is}$ is the rank of feature $s$ in the $i$th permutation; $I(.)$ is the indictor function.

The Gini index does not indicate whether a feature contributes positively or negatively to the prediction of the crosslinked nucleotides or amino acids. To address this limitation, we used a generalized linear regression model using R package Caret. 10-fold cross-validation was used to determine the best parameters ($\alpha = 0.56$ and $\lambda = 0.14$ for prediction of crosslinked nucleotides; $\alpha = 0$ and $\lambda = 7.37$ for prediction of crosslinked amino acids) that maximized the AUC. The coefficient of each feature was used to determine the contribution direction towards the prediction of the crosslinking status.

To evaluate the performance of individual features on prediction, we trained random forest models using each features with Gini decrease value $\geq 1.5$ for crosslinked nucleotide prediction (22 features), and $\geq 2.5$ for crosslinked amino acid prediction (17 features) using the same procedure described above.

## Prediction of crosslinked nucleotides and amino acids using various machine learning models
In addition to the random forest models, we also compared the classification and prediction of crosslinked vs non-crosslinked nucleotides and amino acids using several other machine learning techniques, namely logistic regression, support vector machine (SVM), extreme gradient boosting (XGBoost) trees, and model averaging neural network implemented as avNNet in Caret package[38]. The avNNet method works by using different random number seeds to fit the same neural network model. The model scores from all the resulting trained models are then averaged before being used in predicting the classes[66]. Similar to the random forest model, all the other models were also implemented with SMOTE to address the class imbalance issue, and the model performance was evaluated by 10-fold cross-validation using ROC-AUC scores. The comparative model performance for all the above-mentioned machine learning models is shown in Supplementary Figs. 4 and 9.

## Reporting summary
Further information on research design is available in the Nature Portfolio Reporting Summary linked to this article.

## Data availability
The data supporting the findings of this study are available from the corresponding authors upon request. Protein–RNA complex structures were downloaded from Protein Data Bank (PDB; https://www.rcsb.org). The PDB accession codes of protein–RNA complex structures analyzed in this study were included in Supplementary Datas 3 and 6. The CLIP data analyzed in this study were downloaded from NCBI Sequence Read Archive (https://www.ncbi.nlm.nih.gov/sra) or the ENCODE website (https://www.encodeproject.org) with accession codes summarized in Supplementary Data 3.

## Code availability
The scripts used for this study are available at https://github.com/chaolinzhanglab/PxR3D (https://doi.org/10.5281/zenodo.10602866).

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

## Acknowledgements

The authors thank Judith Kribelbauer for helpful discussion during the early stages of the project. This work was supported in part by grants from the National Institutes of Health (NIH) (R35GM145279 and R03HG009528 to C.Z. and R01GM096889 to X.J.L.). N.D.R. would like to thank Tom Maniatis for mentorship and support. High-performance computation was supported by NIH grants S10OD012351 and S10OD021764.

## Author contributions

Conceptualization and experimental design: H.F. and C.Z. N.D.R: nElavl (HuD) CLIP data generation; Data analysis: H.F., X-J.L, S.M., L.L., D.U., and C.Z., Writing: H.F. and C.Z. All authors critically reviewed the manuscript.

## Competing interests

The authors declare no competing interests.
