## [Peer Review File · Nature Communications]

Structure-based prediction and characterization of photo-crosslinking in native protein-RNA complexesREVIEWER COMMENTS

Reviewer #1 (Remarks to the Author):

In this valuable manuscript, the authors present PxRD3-map: a computational pipeline that analyses structural features associated with photo-crosslinking between protein and RNA in their native complexes. This work has widespread implications in the field of protein-RNA interactions, as UV-crosslinking is used as a standard method of pulling down ribonucleoprotein complexes in high-throughput assays, like CLIP or RNA interaction capture. Despite our awareness that UV-crosslinking has certain limitations, these were not yet systematically evaluated in combination with protein-RNA complex structures. The manuscript addresses this vital issue and is generally well-written; therefore, I would recommend its publication after the authors address the questions and comments below:

1. The authors have analysed amino-acid features that impact protein-RNA crosslinking for unconventional RBPs. A recent study (<https://pubmed.ncbi.nlm.nih.gov/37002329/>) showed that most ucRBPs lack sequence preference, apart from 23 ucRBPs with a clear sequence preference. The authors could include these in their analysis and explore also the nucleotide features contributing to crosslinking in these ucRBPs, taking their findings beyond crosslink selectivity in KH and RRM domains.
2. In the 'PxR3D-map method overview' section, the authors note that they included secondary RNA structure as a nucleotide structural feature in the predictions. However, this is not followed up, presumably because it does not contribute positively to crosslinking. Besides presenting the features that promote crosslinking, the authors might want to discuss the features such as RNA structure that might be negatively associated with crosslinking or have no effect on crosslinking.
3. Could the authors explore the crosslinking associated features for double-stranded RNA binding domains, such as those in the Stauf proteins?
4. The authors evaluated features associated with conventional UV-crosslinking technology; could they expand their analysis to crosslinking with photoreactive nt analogues, especially 4-thiouridine used in PAR-CLIP? Such research would be valuable, as PAR-CLIP method is widely adopted for profiling of protein-RNA interactions. It would be good to understand whether photoreactive nucleotides have different selectivity features, and thus to understand where analysis of PAR-CLIP in combination with CLIP methods that rely on standard UVC crosslinking would be advantageous.
5. Could the authors formulate some suggestions or guidelines for researchers who wish to evaluate crosslinking selectivity from new CLIP data produced for RBPs with unknown crosslinking properties?
6. The study considers that the crosslink sites in identified motifs represent the crosslink sites of the IPed protein. While this likely is the case in most cases of eCLIP data, especially for the canonical RBPs that contain binding domains, it can't be taken for granted as the specificity of CLIP data can be diminished by coIP of other RBPs (and other issues reviewed here <https://www.nature.com/articles/s43586-021-00018-1>). The authors could discuss this as a note of caution, and could consider analysing some motifs in more depth from datasets where the enriched motifs differ between CLIP experiments. For instance, the Fig2 of PMID 36085079 shows that the motifs enriched in eCLIP datasets of PUM1 and IGF2BP1/2 differ considerably from those found in iCLIP or PAR-CLIP (and also from RBNS). Does the analysis of structures in PDBs of such RBPs add any further insight into such discrepancies? Does it provide any insight into which sets of enriched motifs are more consistent with PDBs?
7. Further to the preceding point, the Figure 6 and Table S6 of PMID 36085079 show a large group of RBPs (group 3) that have low agreement with in vitro binding RBNS data. Could the authors now compare the analysis of such datasets with low RBNS agreement with those that have high agreement, to see if 1) if the motifs found in the eCLIP group that overlaps well with RBNS are also

more likely to be found in PDB data, 2) if there are any differences observed when performing some analyses shown in the paper separately on the two sets of datasets? For example, would the prediction performance (as shown in Fig 3a) differ in any way between the two sets?

8. The modelling is limited to the structures of protein RNA complexes that are deposited to PDB. These complexes may represent only a subset of RBP binding motifs; for example, LIN28A has been extensively studied in the context of miRNA processing, and therefore structures with miRNAs dominate the PDB entries. LIN28A also acts by directly binding to mRNAs, where it may recognise additional motifs that may not be as well represented in PDB. Could you assess if there are enriched motifs seen in eCLIP of LIN28A or other RBPs that are not covered by the PDB entries, to understand if the preferences learned by the model, at least at a nucleotide level, could be expanded in the future? Could authors discuss whether increasing the number of structures could broaden our understanding of crosslinking-associated features?

9. Could you explain in more detail the factors that impact sugar puckering chirality? Does a nucleotide have a fixed conformation, or can it change based on RNA context? Can the chirality influence RBP binding to the motif?

Reviewer #2 (Remarks to the Author):

In this study, Feng et al developed a computational method named PxR3D-map to predict UV crosslinking sites in RNA and proteins. CLIP-seq is a commonly used approach to map the binding sites of RNA binding proteins (RBPs). A technical limitation of CLIP-seq is that only a subset of RBP binding sites can be crosslinked and the crosslinking efficiency is variable across RBPs. The authors aimed to understand the photo-crosslinking mechanism/bias by developing a machine learning model.

Overall, the addressed question in this study is important in the RNA field and is difficult as well. The authors performed data mining of published datasets mapping the known crosslinking sites in RNAs and proteins. The random forest approach was used to build the machine learning model and the Gini index scores were used to identify contributing factors. The analyses were well executed. However, the limitation is that the AUROC values are low classifying the crosslinking vs. non-crosslinking sites with the values between 0.69 to 0.80, indicating that the model still cannot explain many differences among the sites. Following are some specific comments.

1. The random forest model was used in this manuscript and I agree that random forest is suitable in this case. Can the authors try a few other machine learning approaches such as logistic regression, support vector machine (SVM), and neuronal network to do the classification and compare the results? Specifically, first, neuronal network can take into account the feature interactions, which may result in better performance (AUROC values) than random forest. Second, since the authors narrowed down a few significant features contributing to the classification using random forest, can they build simpler models just using these relevant features with logistic regression or SVM to do classification? This simplified version can be easier to execute for future users. The analyses can be done to classify crosslinking sites in RNA and proteins, respectively.

2. It would be good if the authors could present AUROC values using individual or a subset of features to classify the crosslinking vs. non-crosslinking sites (RNA and proteins). This will help the readers understand how much improvement we can get by using the machine learning model vs. individual or a subset of features.

3. The model's performance is not so great with the AUROC values between 0.69 and 0.80. The authors discussed the sample size issue. But can there be protein-specific regulation (such as secondary structure) beyond the consensus motif or residual which could not be captured by the modeling approach? The authors can discuss more about the existing limitations and potential future experiments addressing this question.

4. For the RBP binding sites, the authors performed the analyses using one consensus motif for an RBP. But most binding sites can be quite degenerate. Can the authors take that into account by analyzing multiple motifs for an RBP? It would be interesting to see whether the crosslinking preference can lead to biased detection of certain binding sites.

5. For Figure 1b-d, the y-axis (XL freq) needs to be explained.

Reviewer #3 (Remarks to the Author):

Reviewer's comments on the manuscript by Feng et al. "Structure-based prediction and characterization of photo-crosslinking in native protein-RNA complexes"

In the work described in their manuscript, the authors provide a computational approach to predict amino acids in RNA-binding proteins that crosslink to RNA, i.e. PxR3D-map. The overall aim is to merge protein crosslinking and CLIP data. In fact, protein-RNA crosslinking data reveal only a mono- or di/tri-nucleotide moiety, so that the identity of these nucleotides within the overall RNA sequence of the crosslinked RNA remains unknown. Conversely, CLIP data reveal the position of the crosslink within the RNA, but not the protein region, the peptide, or the amino acid that is crosslinked. The authors have therefore developed a prediction tool for identification of crosslinked amino acids in crosslinked RNA-binding proteins. To do this, they used available structures of RNA-binding proteins (taken from the PDB), added available CLIP data for those RNA-protein structures that are in the PDB and applied a computational machine-learning algorithm to identify features relating to exactly how amino acids in the RNA-binding domain interact with the bound RNA.

The authors exemplified and confirmed their approach on three distinct proteins complexed with RNA. They describe distinct features associated with crosslinked nucleotides and they predict crosslinked nucleotides based on the structures. Finally, the authors compared crosslinkability between RRM and KH domains and the amino acids therein. A novel outcome of this is that also cysteine residues are involved in crosslinking, interestingly more frequently in RBPs that do not have a canonical RBD. However, except for the new "spin" of cysteine residues involved in RNA-protein interaction, the work is along the same lines as that described in a paper published recently by Knorlein et al. It aims to achieve the prediction of which amino acids in proteins and their RNA-binding domains are crosslinked to RNA. Both these studies led to the identification of aromatic residues; the study presented here extends its prediction to KH domains and reveals a different amino-acid preference based on 3D structural data.

Overall I regard the approach as a very thorough re investigation/re evaluation of existing datasets (PDB structures where CLIP data are available and recently published RNA-protein crosslinking data). The approach basically summarises, for all the structures investigated, distinct features at the protein and RNA levels. I am, however, not entirely convinced of what is gained from the method (except for a very detailed description of features). It is not clear that the approach allows one to obtain information beyond what already exists. Do MS data not already allow the unambiguous identification of crosslinking sites in a large variety of proteins? In other words: Is the authors' approach able to predict crosslinking sites in proteins where no structural information on the crosslinked protein region is available, but where experiments using CLIP (or one of its derivative methods) reveal crosslinking of a protein? The prediction is biased toward well-characterised RNA-binding domains and very well-defined secondary structure elements of beta sheets and alpha helices, which is to be expected when the algorithm is trained with PDB structures and available CLIP data on these. The MS datasets of Kim and co-workers – and of other groups – show that all amino acids are capable of forming a UV-induced

crosslink to RNA.

Alongside this fundamental question I would like to raise other points that are not entirely clear from the manuscript:

1. The prediction is biased toward well-characterised RNA-binding domains and very well-defined secondary structure elements of beta sheets and alpha helices, as indeed is expected when the algorithm is trained with PDB structures and available CLIP data on these. In large RNA-protein complexes and their respective structures, proteins do not have such well-defined RNA-binding features. It seems that the author used mainly RBPs that have a simply complexity, i.e. one protein-one RNA. Also, protein loop regions and protein regions of low complexity are often involved in RNA-binding/crosslinking. The authors have inferred crosslink sites in the RNA ligand unambiguously for 29 non-redundant 134 protein-RNA complexes representing 25 RBPs. These complexes have a total of 214 nucleotides in the RNA ligands directly contacting the proteins, including 43 nucleotides that were defined as crosslinked nucleotides and the remaining 171 as non-crosslinked nucleotides. These are complexes that interact with (pre-)mRNA. Conversely, the authors have obtained 55 nonredundant complexes (Supplementary Table 6), which consisted of 116 crosslinked amino acids and 1,380 non-crosslinked amino acids that are in direct contact with RNA. The computation prediction resulted in a very interesting and informative table (Supplementary Table 7) also including up- and downstream dipeptides. However, in published large RNA-protein crosslinking datasets lysine residues have been identified to frequently crosslink to RNA. In their analysis (Supp Table 7) lysine residues barely show up. I wonder whether this is due to the selected RNA-protein complexes for machine learning. I wonder whether the authors also included structures of large RNA-protein complexes for machine learning, such as ribosomes and ribosomal proteins, RNA polymerase, spliceosomes, helicases involved in RNA-processing, etc. For these complexes in prokaryotes and in eukaryotes a vast number of highly resolved structures are available. If the authors focus mainly on (pre-)mRNA-protein interactions/crosslinking, then I recommend that they include this important information in the title, the abstract and the introduction of their manuscript. Otherwise, the general nature of these sections, i.e. RNA-protein crosslinking sites in any RNP complex of the manuscript is misleading.
2. With regard to the assignment of crosslinked amino acids within a protein: Does the computational approach exclude additional peptides (or amino-acid sequences) in the protein sequence that cannot be crosslinked? In other words: How would a prediction behave if the entire protein sequence and its corresponding tryptic peptides were given and attempt was made to predict the crosslinked amino acids?

Reviewer #1 (Remarks to the Author):

In this valuable manuscript, the authors present PxRD3-map: a computational pipeline that analyses structural features associated with photo-crosslinking between protein and RNA in their native complexes. This work has widespread implications in the field of protein-RNA interactions, as UV-crosslinking is used as a standard method of pulling down ribonucleoprotein complexes in high-throughput assays, like CLIP or RNA interaction capture. Despite our awareness that UV-crosslinking has certain limitations, these were not yet systematically evaluated in combination with protein-RNA complex structures. The manuscript addresses this vital issue and is generally well-written; therefore, I would recommend its publication after the authors address the questions and comments below:

We are very grateful to Reviewer 1 for underscoring the significance of this study, as well as all the constructive suggestions, which we will address point-by-point below.

1. The authors have analysed amino-acid features that impact protein-RNA crosslinking for unconventional RBPs. A recent study (<https://pubmed.ncbi.nlm.nih.gov/37002329/>) showed that most ucRBPs lack sequence preference, apart from 23 ucRBPs with a clear sequence preference. The authors could include these in their analysis and explore also the nucleotide features contributing to crosslinking in these ucRBPs, taking their findings beyond crosslink selectivity in KH and RRM domains.

We thank the reviewer for this suggestion. Among 23 ucRBPs previously shown to have a sequence preference, we found evidence of protein-RNA crosslinking in three proteins with protein-RNA complex structures available: SERBP1 (6Z6M), ZRANB2 (3G9Y), and RPP25 (6LT7). SERBP1 is a component of the ribosome, which we have analyzed in the revised manuscript in response to Reviewer 3's request (see below). This protein does not have a sufficient resolution in the structure to determine amino acids interacting with crosslinked nucleotides determined by analysis of eCLIP data. Analysis of ZRANB2 and RPP25 is now included in the Discussion (pp 14-15):

“We examined the list of unconventional RBPs previously shown to have clear sequence specificity and found two RBPs with high-resolution protein-RNA complex structures and evidence of crosslinking. Among them, ZRANB2 contains two zinc fingers, with each zinc finger recognizing a 5' splice-site-like AGGUAA motif based on *in vitro* binding^{40, 57} and CLIP data. Our CLIP data analysis also determined that the predominant crosslinking occurs at G2 of this motif, which forms base stacking with an aromatic residue tryptophan (W79; PDB accession: 3G9Y) (Supplementary Fig. 11a). In the other example, RPP25 crosslinks with RNA through a cysteine (C70; PDB accession: 6LT7). However, this amino acid does not directly interact with RNA (Supplementary Fig. 11b).”

Therefore, while the limited availability of data prevents us from systematic structural analysis on unconventional RBPs like canonical RBPs, the results are consistent with our main conclusion that protein-RNA crosslinking can be facilitated by base-stacking with aromatic residues or even transient contacts with cysteine.

2. In the 'PxR3D-map method overview' section, the authors note that they included secondary RNA structure as a nucleotide structural feature in the predictions. However, this is not followed up, presumably because it does not contribute positively to crosslinking. Besides presenting the features that promote crosslinking, the authors might want to discuss the features such as RNA structure that

might be negatively associated with crosslinking or have no effect on crosslinking.

Our analysis demonstrated that the crosslinked nucleotides favor the C2'-endo conformation in their sugar puckers, while the non-crosslinked nucleotides show similar percentages in C2'-endo or C3'-endo conformation, suggesting that the C3'-endo conformation might be negatively associated with crosslinking or have no effects. One limitation of our analysis is that we primarily focused on RBPs binding to single-stranded RNAs, so that we are not effectively evaluating how double-stranded region affect crosslinking, although it is well known that crosslinking of double-stranded RNA has a very low efficiency. This is qualitatively confirmed in our new analysis on a small set of dsRBPs, as described below.

3. Could the authors explore the crosslinking associated features for double-stranded RNA binding domains, such as those in the Staufen proteins?

As suggested by the reviewer, we analyzed the published iCLIP data of Staufen (Sugimoto et al. *Nature* 2015). We found predominant crosslinking of uridine without a sequence-specific motif beyond uridine-rich sequences (Fig. R1 a). Searching for crosslinking in the dsRNA stem sequence GCCAGAA or GCCAGU directly contacting the protein resulted in very few crosslinking events (5 in total) (Fig. 5b). In addition, no crosslinked amino acid was identified in RBS-ID data. These observations are consistent with the original study, which suggested that crosslinking with Staufen frequently occurred upstream of the dsRNA stem region.

Fig. R1: Analysis of protein-RNA crosslink sites for human Staufen. a, Staufen motif identified by de novo motif discovery using mCross. The crosslinking frequencies of different motif positions are shown in the bar plot. b, protein-RNA complex structure (PDB: 6HTU). The double stranded RNA stem directly contacting the protein is highlighted in red (GCCAGGA/GGCAGU).

In addition to Staufen, we also searched for additional dsRBPs with protein-RNA structures and crosslinking data from the literature and the NAKB database (<https://www.nakb.org>). We found four complex structures with evidence of crosslinking in RBS-ID data. In all four cases, the crosslinked amino acids do not appear to contact RNA directly in the stable complex (**Fig. R2**).

Together, our analysis suggests crosslinking between dsRNA and RBPs typically does not occur in dsRNA stem, but flanking single-stranded region, mostly likely through more transient protein-RNA contacts. This is in general consistent with what is previously known. Given that these results are

mostly confirmatory and our searches might not be exhaustive, we decided to provide these results just for reviewers and revisit the topic in a future study.

Fig. R2: protein-RNA complexes for dsRBPs. In each panel, the protein name and PDB accession are indicated. The crosslinked amino acids are highlighted in red.

4. The authors evaluated features associated with conventional UV-crosslinking technology; could they expand their analysis to crosslinking with photoreactive nt analogues, especially 4-thiouridine used in PAR-CLIP? Such research would be valuable, as PAR-CLIP method is widely adopted for profiling of protein-RNA interactions. It would be good to understand whether photoreactive nucleotides have different selectivity features, and thus to understand where analysis of PAR-CLIP in combination with CLIP methods that rely on standard UVC crosslinking would be advantageous.

We agree that it will be interesting to gain structural information of crosslinking that involves photoactive nucleotide analogues such as 4-SU. However, we are not able to include PAR-CLIP data in our analysis because these nucleotide analogues were not used in determination of protein-RNA complex structures. One might try to fine-tune experimental protein-RNA complex structure by inserting 4-SU in silico, which requires substantial expertise and efforts beyond the scope of this study.

5. Could the authors formulate some suggestions or guidelines for researchers who wish to evaluate crosslinking selectivity from new CLIP data produced for RBPs with unknown crosslinking properties?

As we discussed in the manuscript, it will be very important to validate RBP motifs and crosslink sites identified using CLIP using independent assays, such as *in vitro* binding and mutagenesis, or innovative analysis of CLIP data, such as allele-specific binding. We have now restructured our discussion and further elaborated these points in more detail in Discussion (pp 14):

“From these experiences, identification of predominant crosslink sites (especially uridine) outside the expected sequence motif or lack of specific sequence motifs around crosslink sites beyond uridine-rich sequences should signal caution in CLIP data analysis. Importantly, interpretation of protein-RNA interactions mapped by CLIP can benefit from validation using independent approaches such as assessment of binding specificity by *in vitro* binding assays without UV crosslinking^{55, 56}, experimental mutagenesis, or allele-specific binding analyses using CLIP data⁴¹”

6. *The study considers that the crosslink sites in identified motifs represent the crosslink sites of the IPed protein. While this likely is the case in most cases of eCLIP data, especially for the canonical RBPs that contain binding domains, it can't be taken for granted as the specificity of CLIP data can be diminished by coIP of other RBPs (and other issues reviewed here <https://www.nature.com/articles/s43586-021-00018-1>). The authors could discuss this as a note of caution, and could consider analysing some motifs in more depth from datasets where the enriched motifs differ between CLIP experiments. For instance, the Fig2 of PMID 36085079 shows that the motifs enriched in eCLIP datasets of PUM1 and IGF2BP1/2 differ considerably from those found in iCLIP or PAR-CLIP (and also from RBNS). Does the analysis of structures in PDBs of such RBPs add any further insight into such discrepancies? Does it provide any insight into which sets of enriched motifs are more consistent with PDBs?*

This is a valid and important point. To avoid the complication of secondary motifs of co-IP'ed RBPs (or noncanonical mode of protein-RNA interactions), we focused our analysis on RBPs whose primary motif can be unambiguously determined using CLIP data and the motif matches the ligand used in protein-RNA complex structure analysis. For example, while the reviewer referred to the Kuret et al study which identified distinct PUM1 and IGF2BP1/2 motifs, the motif sequences used in this study (UGUANAUA for PUM and CACACCC for IGF2BP) represent the established primary motifs for these RBPs, and our analysis of PUM2 and IGF2BP1 CLIP data resulted in very similar motifs (Feng et al. Mol Cell 2019). The data selection criteria are now clarified in the Method section.

It is an interesting question why motif variants were identified from certain CLIP datasets, and whether this is due to specific CLIP protocols, bioinformatics algorithms, or alternative modes of protein-RNA interactions. Unfortunately, our current PxR3D pipeline can only be applied to RNA sequences used in protein-RNA complex structure determination, so we are unable to analyze other motif variants at the moment.

7. *Further to the preceding point, the Figure 6 and Table S6 of PMID 36085079 show a large group of RBPs (group 3) that have low agreement with in vitro binding RBNS data. Could the authors now compare the analysis of such datasets with low RBNS agreement with those that have high agreement, to see if 1) if the motifs found in the eCLIP group that overlaps well with RBNS are also more likely to be found in PDB data, 2) if there are any differences observed when performing some analyses shown in the paper separately on the two sets of datasets? For example, would the prediction performance (as shown in Fig 3a) differ in any way between the two sets?*

See the response to the point above. It is also worth noting that we previously found RBPs with annotated RBDs are more likely to have reproducible binding-specificity as compared to RBPs without characterized RBDs (Feng et al. Mol Cell 2019). This point is also mentioned in the Discussion.

8. *The modelling is limited to the structures of protein RNA complexes that are deposited to PDB. These complexes may represent only a subset of RBP binding motifs; for example, LIN28A has been extensively studied in the context of miRNA processing, and therefore structures with miRNAs dominate the PDB entries. LIN28A also acts by directly binding to mRNAs, where it may recognise additional motifs that may not be as well represented in PDB. Could you assess if there are enriched motifs seen in eCLIP of LIN28A or other RBPs that are not covered by the PDB entries, to understand if the preferences learned by the model, at least at a nucleotide level, could be expanded in the future? Could authors discuss whether increasing the number of structures could broaden our understanding of crosslinking-associated features?*

We agree that our selection of RBPs is biased towards RBPs with well annotated RNA-binding domains and those binding single-stranded RNA. As for LIN28, we previously performed *de novo* motif discovery using CLIP peaks of LIN28A CLIP data (from mouse ESCs) and LIN28B eCLIP data. Most of the peaks were from mRNAs (Feng et al. *Mol Cell* 2019). We found GGAG and (U)GAU motifs that are recognized by the zinc-knuckle domain and cold shock domain, respectively, suggesting that the binding specificity of LIN28 is shared between miRNAs and mRNAs.

In response to the request from Reviewer 3, we have now performed a separate analysis of the ribosomal proteins (see the last section in Results including new Fig. 7). We found structural features identified in individual protein-RNA complexes can be generalized to large complexes such as the ribosome.

9. *Could you explain in more detail the factors that impact sugar pucker chirality? Does a nucleotide have a fixed conformation, or can it change based on RNA context? Can the chirality influence RBP binding to the motif?*

The RNA sugar pucker conformation is influenced by factors such as hydrogen bond between functional groups on the sugar and neighboring atoms, solvent conditions and so on (Zhang et al. *J. Am. Chem. Soc.* 2012; Auffinger et al. *J. Mol. Biol.* 1997). Pucker dynamics has been observed when these factors are changed. A recent study showed that metal ion binding affects sugar pucker type and balances the kinetic heterogeneity in DNA and RNA tertiary contacts (Steffen et al. *Nat Commun.* 2020).

It has been previously proposed that nucleotide conformation can affect protein-RNA interactions, especially for RRM (ref. 37 in the manuscript). We found sugar pucker tends to differ between crosslinked vs. noncrosslinked nucleotides, although experimental follow-up might be challenging. Without kinetic analysis of protein-RNA complex formation, it is difficult to tell to what extent the sugar conformation can be altered dynamically and/or whether it affects protein-RNA interactions and crosslinking. We acknowledge these limitations, and noted in the Discussion that these observations “await further validation and characterization” (pp 13).

Reviewer #2 (Remarks to the Author):

In this study, Feng et al developed a computational method named PxR3D-map to predict UV crosslinking sites in RNA and proteins. CLIP-seq is a commonly used approach to map the binding sites of RNA binding proteins (RBPs). A technical limitation of CLIP-seq is that only a subset of RBP binding sites can be crosslinked and the crosslinking efficiency is variable across RBPs. The authors

aimed to understand the photo-crosslinking mechanism/bias by developing a machine learning model.

Overall, the addressed question in this study is important in the RNA field and is difficult as well. The authors performed data mining of published datasets mapping the known crosslinking sites in RNAs and proteins. The random forest approach was used to build the machine learning model and the Gini index scores were used to identify contributing factors. The analyses were well executed. However, the limitation is that the AUROC values are low classifying the crosslinking vs. non-crosslinking sites with the values between 0.69 to 0.80, indicating that the model still cannot explain many differences among the sites. Following are some specific comments.

We thank this reviewer for agreeing that this study addressed an important and difficult question and our analyses were well executed. We also appreciate all constructive comments from the reviewer, which we address point-by-point below.

1. The random forest model was used in this manuscript and I agree that random forest is suitable in this case. Can the authors try a few other machine learning approaches such as logistic regression, support vector machine (SVM), and neuronal network to do the classification and compare the results? Specifically, first, neuronal network can take into account the feature interactions, which may result in better performance (AUROC values) than random forest. Second, since the authors narrowed down a few significant features contributing to the classification using random forest, can they build simpler models just using these relevant features with logistic regression or SVM to do classification? This simplified version can be easier to execute for future users. The analyses can be done to classify crosslinking sites in RNA and proteins, respectively.

As suggested by the reviewer, we have now classified crosslinked and noncrosslinked nucleotides/amino acids using different machine learning methods, including logistic regression, support vector machine, XGBoost and neural networks, in addition to random forest. These analyses are now summarized in Supplementary Figs. 3 and 7 and described in relevant sections. All these methods resulted in a very similar performance in the classification tasks, indicating that structure-based classification of crosslink sites is robust. Given the simplicity of random forest in ranking the importance of features for classification, we mainly focused on random forest predictions in our analyses.

Regarding the reviewer's suggestion on feature selection, we are concerned that the current sample size is already quite small, and setting aside a subset of data required for independent feature selection (which should not be used for model training to avoid information leaking) will further worsen the issue. As the number of protein-RNA complex structures and crosslink datasets expand, we will revisit the question in a future study.

2. It would be good if the authors could present AUROC values using individual or a subset of features to classify the crosslinking vs. non-crosslinking sites (RNA and proteins). This will help the readers understand how much improvement we can get by using the machine learning model vs. individual or a subset of features.

Please see our response to the point above.

3. The model's performance is not so great with the AUROC values between 0.69 and 0.80. The authors discussed the sample size issue. But can there be protein-specific regulation (such as secondary

structure) beyond the consensus motif or residual which could not be captured by the modeling approach? The authors can discuss more about the existing limitations and potential future experiments addressing this question.

We acknowledge the limitations of the current analyses, as mentioned by the reviewer above. We have now elaborated these points in the Discussion (pp. 13):

“Altogether, we propose that multiple mechanisms underly the selective photo-crosslinking between protein and RNA, which await further validation and characterization. It is also important to note that our method aims to reveal common structural features shared across many complexes, while protein-specific or rare features contributing to the photocrosslinking efficiency will be missed and they have to be investigated using alternative methods interrogating individual protein-RNA complexes (e.g., refs. ^{23, 24})”

4. For the RBP binding sites, the authors performed the analyses using one consensus motif for an RBP. But most binding sites can be quite degenerate. Can the authors take that into account by analyzing multiple motifs for an RBP? It would be interesting to see whether the crosslinking preference can lead to biased detection of certain binding sites.

Our analysis is limited to RNA ligand sequences used in protein-RNA complex structure determination. However, whenever possible, we did examine whether the degenerate position in the motif affects the identification of protein-RNA crosslink sites. For example, varying the first position NGCAUG for Rbfox does not change the two guanines as the two predominant motif sites. Similarly, the position 5 of the UCUANAUA motif for PUM does not affect the identification of the first uridine or the position further upstream (when it is also a uridine) as the predominant crosslink sites. On the other hand, we also observe cases in which crosslinking preference could bias detection of binding sites. One example is SRSF1, for which we identified a UGGA motif with predominant crosslinking in the first uridine. However, the first uridine does not apparently affect binding specificity based on multiple lines of evidence including allele-specific binding analysis (Feng et al. Mol Cell 2019). The importance of crosslinking bias on interpretation of CLIP data is also elaborated more in the revised Discussion.

See also our response to Reviewer 1 points #5 and #6.

5. For Figure 1b-d, the y-axis (XLfreq) needs to be explained.

Improved as suggested.

Reviewer #3 (Remarks to the Author):

Reviewer's comments on the manuscript by Feng et al. “Structure-based prediction and characterization of photo-crosslinking in native protein–RNA complexes”

In the work described in their manuscript, the authors provide a computational approach to predict amino acids in RNA-binding proteins that crosslink to RNA., i.e. PxR3D-map. The overall aim is to merge protein crosslinking and CLIP data. In fact, protein–RNA crosslinking data reveal only a mono- or di/tri-nucleotide moiety, so that the identity of these nucleotides within the overall RNA sequence of the crosslinked RNA remains unknown.

Conversely, CLIP data reveal the position of the crosslink within the RNA, but not the protein region, the peptide, or the amino acid that is crosslinked. The authors have therefore developed a prediction tool for identification of crosslinked amino acids in crosslinked RNA-binding proteins. To do this, they used available structures of RNA-binding proteins (taken from the PDB), added available CLIP data for those RNA–protein structures that are in the PDB and applied a computational machine-learning algorithm to identify features relating to exactly how amino acids in the RNA-binding domain interact with the bound RNA.

The authors exemplified and confirmed their approach on three distinct proteins complexed with RNA. They describe distinct features associated with crosslinked nucleotides and they predict crosslinked nucleotides based on the structures. Finally, the authors compared crosslinkability between RRM and KH domains and the amino acids therein. A novel outcome of this is that also cysteine residues are involved in crosslinking, interestingly more frequently in RBPs that do not have a canonical RBD.

However, except for the new “spin” of cysteine residues involved in RNA–protein interaction, the work is along the same lines as that described in a paper published recently by Knorlein et al. It aims to achieve the prediction of which amino acids in proteins and their RNA-binding domains are crosslinked to RNA. Both these studies led to the identification of aromatic residues; the study presented here extends its prediction to KH domains and reveals a different amino-acid preference based on 3D structural data.

Overall I regard the approach as a very thorough re investigation/re evaluation of existing datasets (PDB structures where CLIP data are available and recently published RNA–protein crosslinking data). The approach basically summarises, for all the structures investigated, distinct features at the protein and RNA levels. I am, however, not entirely convinced of what is gained from the method (except for a very detailed description of features). It is not clear that the approach allows one to obtain information beyond what already exists. Do MS data not already allow the unambiguous identification of crosslinking sites in a large variety of proteins? In other words: Is the authors' approach able to predict crosslinking sites in proteins where no structural information on the crosslinked protein region is available, but where experiments using CLIP (or one of its derivative methods) reveal crosslinking of a protein? The prediction is biased toward well-characterised RNA-binding domains and very well-defined secondary structure elements of beta sheets and alpha helices, which is to be expected when the algorithm is trained with PDB structures and available CLIP data on these. The MS datasets of Kim and co-workers – and of other groups – show that all amino acids are capable of forming a UV-induced crosslink to RNA.

We thank the reviewer for agreeing that this study is a very thorough analysis by integrating existing datasets of multiple modalities. However, we would like to clarify a few key points regarding the goal of this study and the distinction of our work as compared to Knorlein et al., which was published while this manuscript was being prepared.

First, to clarify, the main goal of this study is to reveal structural features that facilitate UV crosslinking between protein and RNA rather than predicting novel crosslink sites. The mechanistic insights we gain from such analyses is more important from our perspective, since they can help interpret CLIP and interactome capture data and potential bias introduced by crosslinking preference.

Second, while both Knorlein et al. and this study independently determined base stacking between aromatic residues and RNA as one key mechanism of UV crosslinking, the two studies used very

different approaches and also reached different conclusions. The Knorlein et al study started with very detailed experimental dissection of one single protein-RNA complex (RBFOX1 RRM with UGCAUGU), and then they generalized the conclusion by examining additional structures focusing on base stacking. Our study is unbiased from a systematic analysis of all available protein-RNA structures with crosslinking information and provides a methodological framework to make novel discoveries. Thus, the Knorlein study reached a conclusion that most likely over-estimated the contribution of base stacking (up to 78% of crosslink sites). Based on our analyses, base stacking represents one important, but not a predominant, mechanism of protein-RNA photocrosslinking. For example, KH domain is the second most abundant type of RBDs, but they are not crosslinked through base stacking of aromatic residues, but frequently through the glycine (GX or XG dipeptide) upstream of the GXXG motif. In addition, our analyses also revealed the promiscuous crosslinking of cysteine, contribution of certain hydrogen bonds and nucleotide base conformation, although some of the predictions await further validation.

Third, it is important to note that our conclusion on the more promiscuous crosslinking of cysteine during transient protein-RNA contacts is not specific for unconventional RBPs with unusual folds, but also holds for canonical RBDs, such as RRM and KH domains (Fig. 4a-c). Together, these insights could have wide implications in the interpretation of CLIP and interactome capture data.

Alongside this fundamental question I would like to raise other points that are not entirely clear from the manuscript:

1. The prediction is biased toward well-characterised RNA-binding domains and very well-defined secondary structure elements of beta sheets and alpha helices, as indeed is expected when the algorithm is trained with PDB structures and available CLIP data on these. In large RNA-protein complexes and their respective structures, proteins do not have such well-defined RNA-binding features. It seems that the author used mainly RBPs that have a simple complexity, i.e. one protein-one RNA. Also, protein loop regions and protein regions of low complexity are often involved in RNA-binding/crosslinking. The authors have inferred crosslink sites in the RNA ligand unambiguously for 29 non-redundant 134 protein-RNA complexes representing 25 RBPs. These complexes have a 135 total of 214 nucleotides in the RNA ligands directly contacting the proteins, including 43 nucleotides that were 136 defined as crosslinked nucleotides and the remaining 171 as non-crosslinked nucleotides. These are complexes that interact with (pre-)mRNA. Conversely, the authors have obtained 55 nonredundant complexes (Supplementary Table 6), which consisted of 116 crosslinked amino acids and 1,380 non-crosslinked amino acids that are in direct contact with RNA. The computation prediction resulted in a very interesting and informative table (Supplementary Table 7) also including up- and downstream dipeptides. However, in published large RNA-protein crosslinking datasets lysine residues have been identified to frequently crosslink to RNA. In their analysis (Supp Tabel 7) lysine residues barely show up. I wonder whether this is due to the selected RNA-protein complexes for machine learning. I wonder whether the authors also included structures of large RNA-protein complexes for machine learning, such as ribosomes and ribosomal proteins, RNA polymerase, spliceosomes, helicases involved in RNA-processing, etc. For these complexes in prokaryotes and in eukaryotes a vast number of highly resolved structures are available. If the authors focus mainly on (pre-)mRNA-protein interactions/crosslinking, then I recommend that they include this important information in the title, the abstract and the introduction of their manuscript. Otherwise, the general nature of these sections, i.e. RNA-protein crosslinking sites in any RNP complex of the manuscript is misleading.

The reason we decided to focus primarily on relatively simple complexes (i.e., one protein and one RNA) was to ensure the data we analyze has high quality and no potential complications (i.e., crosslinking due to crosslinking of co-IP'd proteins, see point #6 of Reviewer 1's comments).

In response to the reviewer's request, we have now performed a separate analysis of the ribosomal proteins (see the last section in Results of the revised manuscript including new Fig. 7). We found structural features identified in individual protein-RNA complexes can be generalized to large complexes such as the ribosome.

Regarding lysine, our systematic analysis suggests that lysine represents 7.7% of crosslinked amino acids in ribosomal proteins based on RBS-ID data. This proportion is similar to that observed from well characterized RBPs (6.6%).

2. With regard to the assignment of crosslinked amino acids within a protein: Does the computational approach exclude additional peptides (or amino-acid sequences) in the protein sequence that cannot be crosslinked? In other words: How would a prediction behave if the entire protein sequence and its corresponding tryptic peptides were given and attempt was made to predict the crosslinked amino acids?

We limited our prediction of crosslinked vs. noncrosslinked amino acids to those directly interacting with RNA to minimize potential biases. This is because amino acids directly involved in RNA interactions are expected to have unique properties (e.g., more positive charges), as compared to those distal from the protein-RNA interaction interface and such differences can confound prediction of crosslinked vs. non-crosslinked amino acids, as the former are more likely at the protein-RNA interaction interface.

REVIEWERS' COMMENTS

Reviewer #1 (Remarks to the Author):

I thank the reviewers for their informative replies, they have addressed all comments very thoroughly, so I find the manuscript ready for publication.

Reviewer #2 (Remarks to the Author):

The authors addressed most of my comments. I only have minor questions.

1. Regarding the point 2 I raised previously, I am curious to see how well individual features can classify the crosslinking vs. non-crosslinking sites (RNA and proteins). The authors can plot AUROC values using the most important features from their random forest modeling: those with Gini decrease with direction value >1.5 in Figure 3b, value >0.5 in Figure 3d, and value >2.5 in Figure 5b. This will help the readers understand how much improvement a machine learning model can obtain vs. individual features.

2. There is a recent paper from Nature Methods (<https://www.nature.com/articles/s41592-023-02146-w>). Xiao et al developed a method mapping RBP binding sites without UV-crosslinking and IP. The authors can comment on this.

Reviewer #2 (Remarks on code availability):

The README does not have the step-by-step protocol to run the codes. The scripts seem to be rough. The authors can add details, polish the scripts, and describe the steps to execute their command lines.

Reviewer #3 (Remarks to the Author):

Reviewer's comments on the revised manuscript by Feng et al. "Structure-based prediction and characterization of photo-crosslinking in native protein-RNA complexes"

I would like to apologize for the delay in reviewing the revised version of the manuscript by Professor Zhang and colleagues.

The authors responded in a satisfactory manner on my point that I raised on their original version of their manuscript. All in all it is my opinion that the manuscript is in its revised form adds valuable information and a novel AI-based methodology in the field of RNA-protein interactions.

I am happy to state that the manuscript should be published in Nature Communications.